# SUPPRESSING RECENCY BIAS THROUGH IMPLICIT TASK IN TASK-AGNOSTIC CONTINUAL ADAPTATION FOR FOUNDATION LANGUAGE MODELS

## ABSTRACT

Foundation language models have significantly advanced natural language processing but face challenges such as catastrophic forgetting when adapting to dynamic environments with diverse tasks. Recently, among the continual learning (CL) methods for these models, model architecture expansion methods have been spotlighted due to the growth of parameter-efficient fine-tuning (PEFT) methods. However, these methods need to store past PEFT adapters for each task and require task identifiers (task IDs) to distinguish each task, thus limiting their applicability in task-agnostic settings. They also overlook recency bias, where models focus overly on current tasks at the expense of past knowledge. To address these issues, we propose suppressing recency bias (SRB) by using the concept of implicit tasks. SRB assigns a fixed-size adapter to an implicit task, recursively storing historical knowledge through arithmetic operations with current adapters at every time step instead of task IDs. This arithmetic mitigates recency bias by integrating non-overlapping information between historical and current adapters. Our approach requires only simple arithmetic operations without backpropagation, minimizing additional computation, and allocates a fixed-size adapter to the implicit task, resulting in low memory requirements. We evaluate SRB on CL benchmarks for foundational LMs. Experimental results demonstrate that SRB outperforms state-of-the-art methods, achieving superior generalization performance across various task sequences and models by effectively mitigating recency bias.

## 1 INTRODUCTION

Recent advancements in foundation language models (LMs) have demonstrated significant potential in the field of natural language processing (Min et al., 2023; Zhao et al., 2023; Zhou et al., 2023). These models have evolved from pretrained language models (PLMs) (Min et al., 2023) to large language models (LLMs) (Zhao et al., 2023). Early PLMs (Devlin et al., 2019; Liu, 2019; Lewis, 2019) focused on understanding and generating language through tasks like masked language modeling, emphasizing comprehension and generation in text-based applications. Recent LLMs (Achiam et al., 2023; Touvron et al., 2023) have expanded the capabilities of PLMs by increasing the scale of model architectures and training data (Min et al., 2022; Wei et al., 2021; 2022a;b; Yao et al., 2024). This expansion improves generality and adaptability in a variety of tasks. The paradigm of these models involves capturing rich semantic information through pretraining on vast amounts of unlabeled data, followed by fine-tuning to suit specific tasks or domains. This methodology improves performance in various applications and significantly improves the flexibility of the model for different tasks. Despite these advancements, foundation LMs often experience gradual performance degradation when adapting to dynamic environments where a series of tasks from diverse domains are presented (Amba Hombaiah et al., 2021; Dhingra et al., 2022; Jang et al., 2021; Jin et al., 2021; Loureiro et al., 2022; Chen et al., 2023; Cossu et al., 2024; Gupta et al., 2023; Ke et al., 2022). This performance degradation suggests an inherent difficulty for foundation LMs to continuously adapt to multiple environments in a manner similar to human learning processes. A critical challenge in training on a sequence of tasks is catastrophic forgetting, where the model loses previously acquired knowledge when learning new information specific to a task. Addressing catastrophic forgetting requires mechanisms that allow the model to expand and continually adapt to a diverse array of tasks.

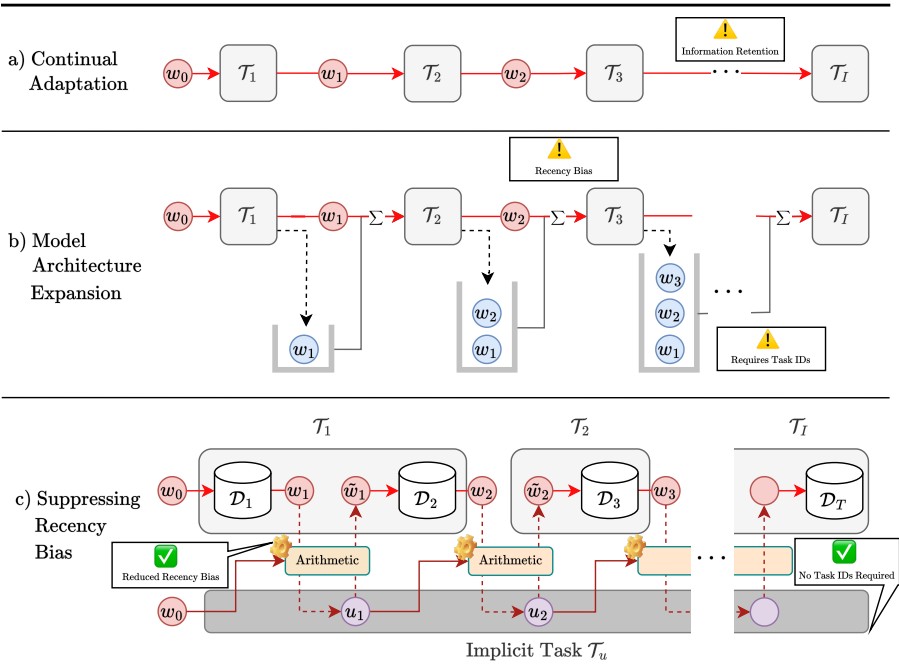

Figure 1: Illustrations of continual adaptation, model architecture expansion, and the proposed suppressing recency bias (SRB) method. (a) Generic continual adaptation sequentially adapts to task series $\mathcal{T}_1, \ldots, \mathcal{T}_I$ using adapter $w_i$. (b) Model architecture expansion methods store adapters corresponding to each past task using task identifiers (task IDs) $i \in \{1, \ldots, I\}$ that distinguish tasks. (c) The proposed SRB method targets the current adapter $w_t$, obtained by optimizing the previous adapter $w_{t-1}$ on a mini-batch dataset $\mathcal{D}_t$ drawn from an unknown task at time step $t$ where the task ID is not provided (Section 3.1). The adapter $u_t$ allocated to the implicit task $\mathcal{T}_u$ is recursively computed via arithmetic operations using $w_t$ and $u_{t-1}$ (Section 3.3). The adapter $\tilde{w}_t$ for the next time step $t + 1$ optimization process is regularized via arithmetic operations to not deviate excessively from $u_t$ (Section 3.4). Detailed arithmetic operations are illustrated in Figure 2.

Continual learning (CL) methodologies have efficiently adapted foundation LMs to downstream tasks while minimizing performance degradation on historical tasks. Inspired by incremental learning patterns observed in the human brain (Constantinescu et al., 2016; Kandel et al., 2000), CL aims for machine learning models to sequentially adapt to a series of tasks while maintaining performance across all tasks. CL approaches for foundation LMs include replay-based methods (Buzzega et al., 2020; Sarfraz et al., 2023; Rebuffi et al., 2017; Zhao et al., 2021; Bang et al., 2021), parameter regularization (Kirkpatrick et al., 2017a; Aljundi et al., 2018; Rongali et al., 2020), and model architecture expansion (MAE) (Aljundi et al., 2017; Hu et al., 2021; Lester et al., 2021; Li & Liang, 2021; Shazeer et al., 2017). Replay-based methods maintain a small buffer that stores portions of observed data from each task to retain past knowledge. However, data storage may not always be feasible due to privacy concerns, and additional computation is required for further learning. Parameter regularization approaches use regularization terms as proxies for the loss values of past domains, determined by distances in the parameter space, to prevent significant deviations from previous parameters. MAE methods dynamically expand the network architecture to integrate new information in a CL manner (Gururangan et al., 2021; Wistuba et al., 2023).

Recently, as parameter-efficient fine-tuning (PEFT) has become the standard approach to continual adaptation, MAE methods have gained attention (Dettmers et al., 2024; Wang et al., 2023; Wu et al., 2024; Yan et al., 2023). MAE strategy stores PEFT adapters for each task and combines the outputs of past and current adapters to update the model. This approach has demonstrated superior retention of past knowledge compared to existing methods by storing and freezing adapters during adaptation (Zhang et al., 2023a; Wang et al., 2023). Despite these successes, MAE strategies require task

identifiers (task IDs) to store the adapter corresponding to each task, making them difficult to apply in *task-agnostic scenarios* (Criado et al., 2022; Pentina & Lampert, 2015). Moreover, this strategy does not address the issue of *recency bias* (Ray, 2023), where excessive focus on the current task leads to the loss of past knowledge (Peysakhovich & Lerer, 2023). This recency bias problem is exacerbated in continual adaptation settings, where the model repeatedly learns about the current task (Criado et al., 2022; Pentina & Lampert, 2015).

To address these challenges, we propose a method called *suppressing recency bias* (SRB), which introduces an implicit task and assigns adapters to the task, thereby eliminating the need for task IDs and reducing redundant information acquisition (see Figure 1). We focus on the current adapter, trained on a mini-batch dataset drawn from a task without a task ID. This adapter is recursively integrated into an implicit task adapter over time to construct historical knowledge, utilizing arithmetic operations. These operations are designed to compare the historical knowledge with the current information to suppress repetitive information before storing it in the implicit task adapter. Finally, we modify the current adapter by regularizing it from deviating excessively from the implicit task adapter. The advantages of SRB are as follows:

- SRB can be applied in task-agnostic settings and excels at adapting to each task while preserving historical knowledge by reducing recency bias.
- Implicit tasks require only arithmetic operations that do not necessitate backpropagation, minimizing additional computation.
- SRB allocates only a fixed-size adapter to the implicit task, resulting in low additional memory requirements.

We compare the proposed method with state-of-the-art techniques on CL benchmarks for foundation LMs in task-agnostic continual adaptation. The proposed method demonstrates superior generalization performance over existing methods across task series of various orders, lengths, and models. We show that our method's enhanced generalization performance is achieved by reducing the loss of past knowledge due to recency bias observed in existing methods.

## 2 PRELIMINARIES

### 2.1 CONTINUAL ADAPTATION FOR FOUNDATION LMS

**Continual Adaptation** CL has been a long-standing challenge in machine learning (McCloskey & Cohen, 1989). In a CL setting, a model sequentially adapts to tasks $\mathcal{T}_i$ for each task ID $i \in \{1, \dots, I\}$. We denote the dataset assigned to task $\mathcal{T}_i$, consisting of $N$ samples, as $\mathcal{D}_i = \{(\boldsymbol{x}_n, \boldsymbol{y}_n) : n = 1, \dots, N\}$, where $\boldsymbol{x}_n$ is the input text and $\boldsymbol{y}_n$ is the corresponding target text. Before starting continual adaptation, the model is initialized with weights $W_0 \in \mathbb{R}^D$ of dimension $D$ from a foundation LM. The adaptation objective at each time step is defined as:

$$L(W_{i-1}, \mathcal{D}_i) = \frac{1}{N} \sum_{(\boldsymbol{y}_n, \boldsymbol{x}_n) \in \mathcal{D}_i} \log p(\boldsymbol{y}_n | \boldsymbol{x}_n; W_{i-1}), \tag{1}$$

where $p(\boldsymbol{y}_n \mid \boldsymbol{x}_n; W_{i-1})$ is the probability of generating $\boldsymbol{y}_n$ given $\boldsymbol{x}_n$ using the model weights from the previous time step $W_{i-1}$. The updated weights $W_i$ are then computed by optimizing the adaptation objective:

$$W_i \leftarrow \underset{W_{i-1}}{\arg\max}\, L(W_{i-1}, \mathcal{D}_i). \tag{2}$$

However, this sequential learning approach risks losing past knowledge because it relies solely on the previous weights $W_{i-1}$, making it susceptible to catastrophic forgetting.

**Continual Learning for Foundation LMs** To mitigate catastrophic forgetting, replay-based methods that store and continually utilize past data have been employed (Buzzega et al., 2020; Sarfraz et al., 2023; Rebuffi et al., 2017; Zhao et al., 2021; Bang et al., 2021). These methods maintain a memory buffer containing data from previous tasks, allowing the model to reference prior information and alleviate the loss of past knowledge. However, replay-based methods can be impractical in real-world applications due to privacy concerns that make storing past task data unrealistic. In addition, they require extra computation to train on the data in the memory buffer.

| Semantic Intent | Arithmetic Operation |
|---|---|
| Multi-task learning | $\tau_\alpha + \tau_\beta$ |
| Unlearning | $\tau_\alpha - \tau_\beta$ |
| Domain transfer | $\tau_\gamma + (\tau_\alpha - \tau_\beta)$ |

Table 1: Semantic intent and their arithmetic operation for $\alpha, \beta$ and $\gamma$ tasks.

Alternatively, parameter regularization methods have been explored, which save previous weights and continuously access them during adaptation to preserve historical knowledge (Kirkpatrick et al., 2017a; Aljundi et al., 2018; Rongali et al., 2020). These methods introduce a regularization loss that prevents current weights from deviating significantly from past weights. Specifically, L2 regularization helps prevent the weights from becoming excessively large, resulting in improved performance (Zhang et al., 2023c; Lin et al., 2022).

## 2.2 CONTINUAL ADAPTATION USING PARAMETER-EFFICIENT FINE-TUNING

**Parameter-efficient Fine-tuning** The PEFT methods propose inserting a adapter weight $w_i \in \mathbb{R}^d$ of dimension $d$ at various positions in the Transformer (Vaswani, 2017) architecture commonly used in foundation LMs, such as after attention and feedforward networks (Houlsby et al., 2019; Li & Liang, 2021; He et al., 2021). Continual adaptation through the PEFT approach is performed by updating the adapter as follows:

$$w_{i+1} \leftarrow \arg\max_{w_i} L(\{w_i, W_0\}, \mathcal{D}_{i+1}), \tag{3}$$

where $W_0$ represents the fixed weights of the foundation LMs and only $w_i$ are updated. One of the most effective PEFT methods is a low-rank adaptation (LoRA) (Hu et al., 2021), which has gained significant attention and has become a standard approach for adapting LLMs such as LLaMA (Touvron et al., 2023) under limited computational resources. LoRA decomposes the adapters by mapping the input vector to a lower-dimensional space and then back to the original dimension. Specifically, for dimensions $k$ and $l$, given an input $\boldsymbol{z} \in \mathbb{R}^k$ and output $\boldsymbol{h} \in \mathbb{R}^l$ in the Transformer, LoRA modifies $\boldsymbol{h}$ as:

$$\boldsymbol{h} \leftarrow \boldsymbol{h} + BA\boldsymbol{z}, \tag{4}$$

where $A \in \mathbb{R}^{r \times k}$ and $B \in \mathbb{R}^{l \times r}$ are projection matrices, with rank $r$ much smaller than $\min(l, k)$. Here, $d = lk$ denotes the dimensionality of the adapter weight $w_i = B_i A_i$. LoRA can be applied to any weight matrix but is typically used in query and value projection matrices (Hu et al., 2021). The matrix $A$ is initialized from a Gaussian distribution, while $B$ is initialized to zeros to allow recovery of $W_0$. During adaptation, only the adapter weights are updated. Since $d$ is much smaller than $D$, most of the model weights remain identical to $W_0$. Similar to parameter regularization approaches, this characteristic of PEFT helps preserve past knowledge by preventing the current weights from deviating too far from their previous weights.

**Model Architecture Expansion** As the adoption of LoRA as a standard method, MAE techniques that expand adapters as tasks increase have gained attention (Dettmers et al., 2024; Wang et al., 2023; Wu et al., 2024; Yan et al., 2023). For the current task $i$, these methods modify $\boldsymbol{h}$ using the LoRA weights $w = BA$ as follows:

$$\boldsymbol{h} \leftarrow \boldsymbol{h} + (w + w_1 + w_2 + \cdots + w_{i-1})\boldsymbol{z}, \tag{5}$$

where $w_j = B_j A_j$ for $j = 1, \ldots, i - 1$ are the adapters for past tasks, which are stored and kept frozen after past adaptation. The outputs of all adapters are summed to modify $\boldsymbol{h}$, effectively integrating knowledge from past and current tasks. This process aims to prevent the current adapter from forgetting historical knowledge by referencing the outputs of the stored adapters during learning (Zhang et al., 2023a; Wang et al., 2023).

## 2.3 ARITHMETIC OPERATIONS OF TASK VECTORS FOR SEMANTIC OPERATIONS

Recent studies have demonstrated that arithmetic operations between adapted weights can concretely implement semantic intents (Ilharco et al., 2022). These semantic intents include improving performance of downstream task, alleviating biases or unwanted behaviors, aligning the model with human

preferences, or updating the model with new information. Such semantic intents are based on the concept of a task vector. The task vector is defined as:

$$\tau = W_i - W_0, \tag{6}$$

where $W_i$ represents the adapted weights for task $i$, and $W_0$ denotes the initial weight of the foundation LM. This approach encodes the information needed to adapt to a specific task, introducing a new paradigm for neural network editing. Inspired by studies on weight interpolation (Guo et al., 2023; Wortsman et al., 2022; Rame et al., 2022; 2024), task vectors enable task arithmetic, performing element-wise operations to edit various models. For example, adding task vectors can enhance multi-task model performance to achieve generalized capabilities (first row in Table 1), while unlearning can help the model remove unwanted behaviors or forget specific tasks (second row in Table 1). Furthermore, when tasks share similar relationships, combining task vectors allows concrete computations of abstract concepts such as domain transfer (third row in Table 1).

## 3 SUPPRESSING RECENCY BIAS

### 3.1 OVERALL PROCESS

In **task-agnostic continual adaptation**, task IDs are not provided, and the model continually adapts without explicit knowledge of task boundaries. This scenario differs from the standard continual adaptation setting (as described in Eq. (3)), where datasets $\mathcal{D}_i$ are associated with specific tasks. Instead, we consider mini-batches of data $\mathcal{D}_t$ of size $B$ at each time step $t \in [1, T]$, where $T$ is the total number of time steps. The optimization process at each time step is defined as:

$$w_t \leftarrow \underset{\tilde{w}_{t-1}}{\arg\max} \, L(\{\tilde{w}_{t-1}, W_0\}, \mathcal{D}_t), \tag{7}$$

where $w_t$ represents the updated adapter weights at time $t$, $\tilde{w}_0$ is assigned as the zero-initialized $w_0$, and $W_0$ denotes the initial weights of the foundation LMs. The objective function $L$ is the log-likelihood defined as:

$$L(\{\tilde{w}_{t-1}, W_0\}, \mathcal{D}_t) = \frac{1}{B} \sum_{(\boldsymbol{x}_n, \boldsymbol{y}_n) \in \mathcal{D}_t} \log p(\boldsymbol{y}_n \mid \boldsymbol{x}_n; \{\tilde{w}_{t-1}, W_0\}), \tag{8}$$

where $(\boldsymbol{x}_n, \boldsymbol{y}_n)$ are input-output text pairs in the mini-batch $\mathcal{D}_t$. In this setting, the union of all datasets over tasks is equivalent to the union over time steps, $\bigcup_i \mathcal{D}_i = \bigcup_t \mathcal{D}_t$. This implies that the data is presented sequentially over time without explicit task boundaries. To compute $\tilde{w}_t$, we introduce a recursive arithmetic operation defined as:

$$\tilde{w}_t, u_t \leftarrow \text{Arithmetic}(w_t, u_{t-1}), \tag{9}$$

where $u_t \in \mathbb{R}^d$ is an adapter allocated to an **implicit task** $\mathcal{T}_u$, having the same dimensionality as $w_t$. The function Arithmetic($\cdot$) performs recursive operations with $u_0$ initialized as $w_0$ and then modifies the $w_t$. This function allows the implicit task to incorporate information from past data without storing adapters for each task individually, thereby operating in a **task-agnostic manner**. The step-by-step implementation is provided in Algorithm 1.

### 3.2 PROBLEM STATEMENT

Suppose that we define the arithmetic operation as $\tilde{w}_t = w_t + u_{t-1}$ with $u_{t-1} = w_1 + \cdots + w_{t-1}$. In that case, this operation corresponds to the arithmetic used in Eq. (5). This method effectively accumulates the adapters from all previous time steps, analogous to the arithmetic used in MAE approaches representing multi-task learning (as shown in Table 1). However, this approach treats each task independently without considering the sequential relationships or redundancies between adapters. Consequently, it cannot prevent the duplication of information across adapters. This mechanism is vulnerable to **catastrophic forgetting** due to **recency bias**, where the model disproportionately focuses on recent data at the expense of past knowledge. This issue is particularly pronounced in continual adaptation scenarios where mini-batches drawn from the same task are repeatedly presented, and subsequent tasks are introduced sequentially. The accumulation of redundant information and the lack of mechanisms to mitigate recency bias lead to inefficient learning and degradation of performance on previous tasks.

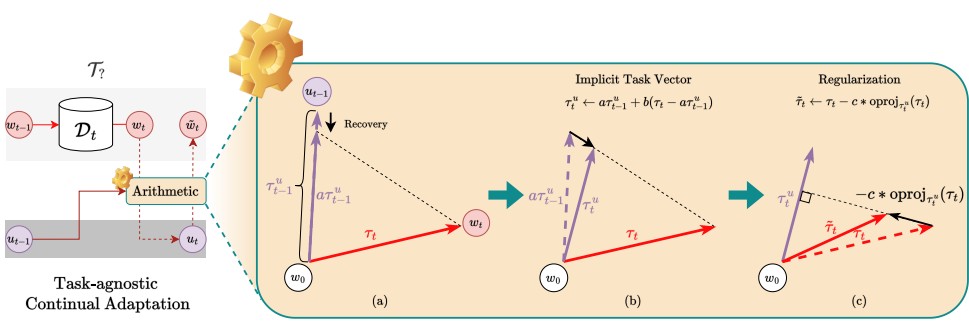

Figure 2: Illustration of arithmetic operations in SRB.

### 3.3 ARITHMETIC FOR IMPLICIT TASK VECTORS

In the implicit task, we aim to integrate historical knowledge with current information to generalize across multiple tasks and suppress recency bias. According to Appendix A.1, the zero-initialized adapter corresponds to a task vector, so the task vector for the current task is $\tau_t = w_t - w_0$ and the task vector for the implicit task is $\tau_t^u = u_t - w_0$. Task vector arithmetic is based on the notion that generalized weights exist in the interpolation regions between weights (Guo et al., 2023; Wortsman et al., 2022; Rame et al., 2022; 2024). Therefore, we compute the implicit task vector through interpolation of $w_0$, $u_{t-1}$, and $w_t$ as follows:

$$u_t = \lambda_1 w_0 + \lambda_2 u_{t-1} + \lambda_3 w_t, \tag{10}$$

subject to the constraints $\lambda_1 + \lambda_2 + \lambda_3 = 1$ and $0 \leq \lambda_1, \lambda_2, \lambda_3 \leq 1$. Since $\lambda_1 = 1 - \lambda_2 - \lambda_3$, we can rewrite the equation as $u_t - w_0 = \lambda_2(u_{t-1} - w_0) + \lambda_3(w_t - w_0)$. Expressing this in terms of the task vectors $\tau_t$ and $\tau_t^u$, we have $\tau_t^u = \lambda_2 \tau_{t-1}^u + \lambda_3 \tau_t$. By setting $\lambda_2 = a(1 - b)$ and $\lambda_3 = b$, for $0 \leq a, b \leq 1$, the implicit task vector becomes

$$\tau_t^u = a\tau_{t-1}^u + b(\tau_t - a\tau_{t-1}^u). \tag{11}$$

This result is a weighted version of the domain transfer in Table 1. The hyperparameter $a$ controls the influence of the previous implicit task vector $\tau_{t-1}^u$. The value of $a$ close to 0 effectively emphasize the impact of the foundation LM weights $W_0$ (see Figure 2 (a)). In contrast, the value of $a$ close to 1 retains more historical information. The hyperparameter $b$ determines the degree to which new information is incorporated. The value of $b$ close to 0 causes the model to respond slowly to rapidly changing information, acting as a low-pass filter (see Figure 2 (b)). The term $(\tau_t - a\tau_{t-1}^u)$ compares the current task vector $\tau_t$ with the previous scaled implicit task vector $a\tau_{t-1}^u$. This difference captures the new distinctive information not already represented in the past knowledge. By adding this adjusted difference to $a\tau_{t-1}^u$, we effectively reduce **duplicated information** and prevent excessive growth of redundant task information in the implicit task.

### 3.4 ARITHMETIC FOR REGULARIZATION

We leverage the non-overlapping information from the implicit task adapter to modify the current adapter. However, sufficient diversity among weights is necessary for generalized weights to exist in the interpolation regions between weights (Wortsman et al., 2022; Rame et al., 2022). Paradoxically, the implicit task vector acts as a low-pass filter, limiting diversity. Therefore, we ensure that the current task vector is sufficiently distant from the implicit task vector to ensure diversity. To achieve this, we calculate a regularization term that exerts a stronger attractive force on the current task vector $\tau_t$ when it is less similar to the implicit task vector $\tau_t^u$, using task vector arithmetic. First, we compute the orthogonal projection of $\tau_t$ onto $\tau_t^u$:

$$\text{oproj}_{\tau_t^u}(\tau_t) = \tau_t - \frac{\tau_t \cdot \tau_t^u}{\tau_t^u \cdot \tau_t^u} \tau_t^u, \tag{12}$$

where $\cdot$ denotes the dot product. This operation calculates the component of $\tau_t$ that is orthogonal to $\tau_t^u$, which effectively implies the dissimilarity between the two vectors. As illustrated in Figure 2 (c), the magnitude of this orthogonal component is proportional to the angle between the two vectors.

Therefore, the attractive force towards the implicit task vector is dynamically adjusted according to the similarity between $\tau_t$ and $\tau_t^u$. Finally, we modify $\tau_t$ using this attraction vector:

$$\tilde{\tau}_t \leftarrow \tau_t - c \cdot \text{oproj}_{\tau_t^u}(\tau_t), \tag{13}$$

where $0 \leq c \leq 1$ is a hyperparameter that controls the strength of the attraction. This operation dynamically regularizes the current task vector by applying a stronger attraction when $\tau_t$ is less similar to $\tau_t^u$ and a weaker attraction when they are more similar. This regularizing operation effectively balances **increasing diversity** and **reducing recency bias** by using the implicit task vector as a support. The diversity enhances the autonomy of the current task vector, thus maintaining adaptability to each task. The final modified $\tilde{\tau}_t$ is equivalent to $\tilde{w}_t$ and is used in the subsequent optimization step in Eq. (7).

## 4 EXPERIMENTS

### 4.1 EXPERIMENTAL SETUP

**Datasets and Metric** We evaluated our approach using a *standard CL benchmark* designed for foundation LMs (Qin & Joty, 2021). This benchmark consists of four text classification datasets introduced by Zhang et al. (2015): AG News, Amazon Reviews, DBpedia, and Yahoo Answers. Following the previous study (Qin & Joty, 2021), we applied three different orders of CL settings to these datasets (Order 1, 2, and 3). To assess performance on longer task sequences, we conducted experiments on a *long CL benchmark* comprising 15 datasets (Razdaibiedina et al., 2023). This extended benchmark includes the initial four CL benchmarks along with the Yelp reviews (Zhang et al., 2015), four tasks from the GLUE benchmark (MNLI, QQP, RTE, SST2) (Wang, 2018), five tasks from the SuperGLUE benchmark (WiC, CB, COPA, MultiRC, BoolQ) (Wang et al., 2019), and the IMDB movie reviews dataset (Maas et al., 2011) (Order 4, 5, and 6). Sequences of tasks are provided in Appendix C.1. Following previous work (Razdaibiedina et al., 2023), we randomly selected 1,000 samples per task for training and reserved 500 samples per class for validation. For evaluation metrics, we used accuracy (Chaudhry et al., 2018) and reported the average accuracy (Avg.) for all tasks after training on the last task.

**Comparison Methods** We compared SRB with seven other CL methods for foundation LMs. EWC (Kirkpatrick et al., 2017b) employs a regularization loss based on the Fisher information matrix to prevent significant weight updates that could interfere with previously learned tasks while fine-tuning the entire model. Replay (Buzzega et al., 2020) uses a memory buffer containing data from previous tasks to fine-tune the whole model, retraining on samples from previous tasks when learning new ones to avoid forgetting. Learning without Forgetting (LwF) (Li & Hoiem, 2017) adds a regularization loss before learning a new task to ensure that the shared representation layers remain similar to those of previous representations. LoRA (Hu et al., 2021) learns a series of tasks using fixed-size LoRA adapters without retraining on samples of earlier tasks or employing regularization. Incremental low-rank adaptation (IncLoRA) (Zhang et al., 2023a) incrementally adds new LoRA adapters for each task in a series, similar to LoRA, but without retraining on previous task samples or using regularization. Orthogonal low-rank adaptation (O-IncLoRA) (Wang et al., 2023) builds upon IncLoRA by introducing an additional loss that enforces orthogonality among the adapters stored for each task. L2 (Zhang et al., 2023c) applies an L2 regularization loss to constrain the LoRA adapters from significantly changing while learning new tasks. Additionally, We referred to the multitask learning baseline (MTL) as the upper bound and per-task fine-tuning (PerTaskFT) from (Du et al., 2024) for the benchmark. Further details of the experimental setup are provided in Appendix C.2.

**Implementation Details** We adopted two open-source foundation LMs in line with previous studies: the encoder-decoder T5 (Raffel et al., 2020) and the decoder-only LLaMA (Touvron et al., 2023). We adopted the large version of the T5 model. For LLaMA, we employed the latest 8B parameter version, LLaMA3 (Dubey et al., 2024), and LLaMA3-chat, which includes additional instruction tuning performed on the base model. We adhered to their official implementations for all comparison methods and followed the hyperparameters reported in the original papers to ensure consistency with existing CL benchmarks. We used the AdamW optimizer (Loshchilov, 2017) with $\beta_1 = 0.9$ and $\beta_2 = 0.999$ and the batch size was 64. For our SRB method, the hyperparameters $(a, b, c)$ were uniformly set to $(0.99, 0.025, 0.15)$ across all experiments. We set the learning rates to 0.001 and

Table 2: Accuracy (%) of each order and Average accuracy (Avg., %) on the standard CL benchmark (Order 1, 2 and 3) and the long CL benchmark (Order 4, 5 and 6) for the T5 model. All results are averaged over three runs. * indicates performance results from (Du et al., 2024), ✓ of Expan. indicates the MAE method, and ✓ of Task IDs denotes the task-agnostic settings.

| Method | Order | | | Avg. | Order | | | Avg. | Expan. | Task IDs |
|---|---|---|---|---|---|---|---|---|---|---|
| | 1 | 2 | 3 | | 4 | 5 | 6 | | | |
| PerTaskFT* | 70.0 | 70.0 | 70.0 | 70.0 | 78.1 | 78.1 | 78.1 | 78.1 | - | |
| MTL* | 80.0 | 80.0 | 80.0 | 80.0 | 76.5 | 76.5 | 76.5 | 76.5 | - | |
| EWC* | 48.7 | 47.7 | 54.5 | 50.3 | 45.3 | 44.5 | 45.6 | 45.1 | - | |
| Replay* | 55.2 | 56.9 | 61.3 | 57.8 | 55.0 | 54.6 | 53.1 | 54.2 | - | |
| LwF* | 54.4 | 53.1 | 49.6 | 52.4 | 50.1 | 43.1 | 47.4 | 46.9 | - | ✓ |
| IncLoRA | 71.4 | 66.2 | 70.7 | 69.4 | 62.3 | 66.2 | 63.5 | 64.0 | ✓ | |
| O-IncLoRA | 77.1 | 76.2 | 76.6 | 76.6 | 68.4 | 68.8 | 71.4 | 69.5 | ✓ | |
| LoRA | 61.9 | 62.1 | 68.8 | 64.3 | 53.7 | 44.4 | 39.8 | 46.0 | - | |
| L2 | 66.0 | 63.0 | 63.9 | 64.3 | 49.1 | 46.9 | 12.8 | 36.3 | - | - |
| SRB | **78.1** | **78.2** | **77.5** | **77.9** | **70.5** | **71.4** | **73.3** | **71.7** | - | |

adopted LoRA as the PEFT method for the adapter weights. All experimental results are reported as the average over three runs.

## 4.2 RESULTS OF CL BENCHMARK

Table 2 presents the average accuracy of CL methods for foundation LMs on both the standard and long CL benchmarks. The MAE approaches, specifically IncLoRA and O-IncLoRA, demonstrated superior performance compared to traditional CL methods, with O-IncLoRA achieving the highest accuracy. In contrast, task-agnostic settings that do not utilize task IDs, such as those employing LoRA and L2, generally exhibited comparable or decreased performance on the long CL benchmark relative to existing methods. These findings suggest that reusing adapters allocated to each task effectively enhanced performance and that enforcing orthogonality among adapters, as in O-IncLoRA, was beneficial for further improvement. Our proposed SRB method attained even higher performance than O-IncLoRA despite operating without task IDs, whereas O-IncLoRA relied on them. In the task-agnostic settings, SRB achieved significant performance gains of approximately 21% and 54% compared to LoRA on the standard and long CL benchmarks, respectively.

## 4.3 RESULTS OF CL BENCHMARK FOR LLM

Table 3 shows the average accuracy of LoRA, MAE methods, and our proposed SRB on the standard CL benchmark using the LLaMA3 models. The LLaMA3 and LLaMA3-chat models generally exhibited more stable and higher performance across various methods than the T5 model. However, IncLoRA and O-IncLoRA, which displayed strong performance on the T5 model, performed similarly to or slightly worse than LoRA, even when task IDs were provided. In contrast, SRB recorded higher performance than LoRA in this setting. These observations imply that SRB consistently delivers superior performance across different foundation LMs. Moreover, the unique feature of SRB in suppressing recency bias appeared to be a significant factor in enhancing performance for the LLaMA3 models.

Table 3: Accuracy (%) of each order and average accuracy (Avg., %) on the standard CL benchmark (Order 1, 2 and 3) for LLaMA3 and LLaMA3-chat. All results are averaged over 3 runs.

| Model | Method | Order | | | Avg. |
|---|---|---|---|---|---|
| | | 1 | 2 | 3 | |
| LLaMA3 | LoRA | 75.8 | 75.9 | 74.4 | 75.4 |
| | IncLoRA | 75.0 | 75.2 | 75.7 | 75.3 |
| | O-IncLoRA | 74.6 | 74.7 | 74.8 | 74.7 |
| | SRB | **79.0** | **80.5** | **77.0** | **78.8** |
| LLaMA3-chat | LoRA | 75.6 | 75.6 | 75.7 | 75.6 |
| | IncLoRA | 75.1 | 74.9 | 76.2 | 75.4 |
| | O-IncLoRA | 74.6 | 74.5 | 74.9 | 74.7 |
| | SRB | **78.9** | **80.3** | **78.0** | **79.1** |

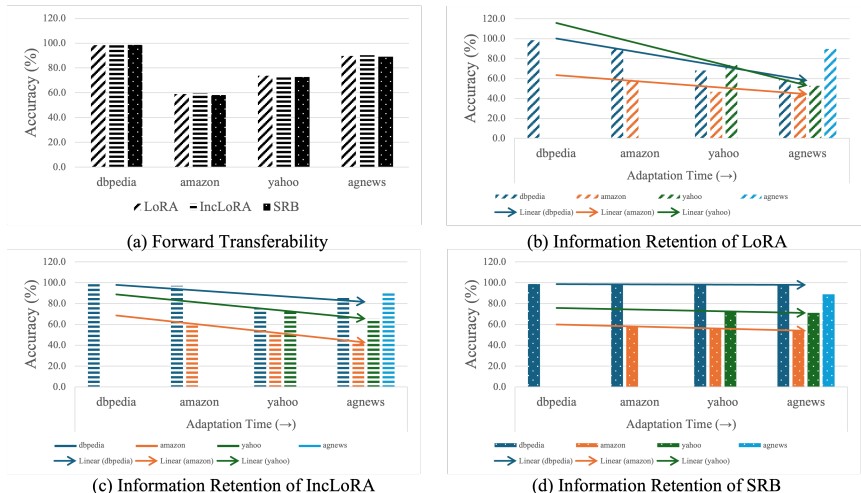

Figure 3: (a) Accuracy (%) of each current task after adaptation in Order 1 using the T5 model. (b)–(d) Accuracy of each method along with adaptation time.

## 5 DISCUSSIONS

In this section, we discuss the challenges of recency bias, analyze the role of diversity and evaluate efficiency. Detailed discussions, including ablation studies, can be found in Appendix E.

### 5.1 ANALYSIS OF RECENCY BIAS

We analyzed the recency bias by comparing the forward transferability, which refers to a capacity to leverage learned knowledge from previous tasks to enhance performance on current task, and the ability to preserve historical knowledge of SRB with existing methods. Figure 3 illustrates the forward transferability of each method on individual tasks (Figure 3 (a)) and the preservation of previous information over adaptation time (Figures 3 (b)–(d)) on the standard CL benchmark. As shown in Figure 3 (a), LoRA and IncLoRA exhibited slightly higher or comparable performance compared to SRB when adapting to each target task. However, Figures 3 (b) and 3 (c) revealed that, as adaptation progresses over time, both of these methods experienced a rapid decline in performance on previously adapted tasks (indicated by the dotted lines). This outcome demonstrates that existing methods are prone to catastrophic forgetting due to excessive adaptation to the latest task, highlighting the issue of recency bias. In contrast, Figure 3 (d) shows that SRB maintained nearly parallel dotted lines over time, indicating that the performance on past tasks remained largely preserved despite ongoing adaptation. These results suggest that the implicit task and the arithmetic operations designed to suppress recency bias in SRB effectively mitigated the forgetting of past information.

### 5.2 ANALYSIS OF DIVERSITY

We analyzed how variations in the hyperparameter $c$, which controls the diversity of the current task vector during optimization, affect SRB. We measured diversity as $\log(1 - s)$ (Lee & Chang, 2024), where $s$ is the cosine similarity between the implicit and current task vectors. Figure 4(a) illustrates that increasing the value of $c$ led to a decrease in diversity over the adaptation time. This outcome indicates that the regularization imposed by $c$ effectively reduced the diversity. Figure 4(b) shows that the performance improved as diversity decreased until $c = 0.75$. Specifically, the model achieved stable performance when $c$ was set between 0.05 and 0.5. However, when $c$ was set to 1.0, diversity rapidly declined from the 10th task onward. For tasks 13 to 15, the diversity measure became undefined (indicated by triangle points in Figure 4), resulting in an average accuracy of 0.0%. This result suggests that the optimization process failed to produce sufficiently diverse adapter without adequate regularization. Consequently, the model ability to generalize deteriorates, leading to a significant decrease in performance similar to the findings in (Rame et al., 2022).

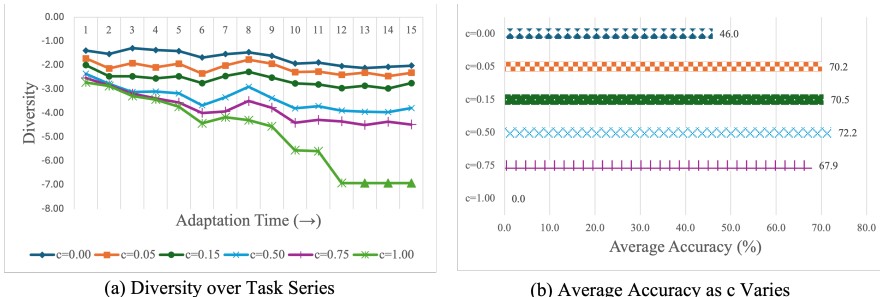

(a) Diversity over Task Series    (b) Average Accuracy as c Varies

Figure 4: (a) Diversity over task series for various values of $c$ in Order 4 for T5. (b) Average accuracy (%) for different values for each value of $c$.

## 5.3 ANALYSIS OF COMPUTATIONAL EFFICIENCY

We compared and analyzed the efficiency of SRB with existing methods. Table 4 summarizes the computational resources required by each approach. IncLoRA and O-IncLoRA required more memory and computational time than LoRA, increasing training times. Specifically, IncLoRA maintains multiple adapters, increasing memory usage and computational overhead during the forward pass as it processes each adapter separately. O-IncLoRA further introduces an additional loss term to enforce orthogonality among

Table 4: Comparison of computational and memory efficiency for each method, including average accuracy (Avg., %) across all tasks in Order 1. #Adapters denotes the number of adapters needed when processing the $I$-th task. #Forward and #Backward indicate the computational load required to process on one mini-batch containing $B$ samples during the forward and backward passes, respectively. Elapsed Time indicates that the time taken to adapt the last task in Order 1 for the T5 model.

| Method | Avg. | #Adapter | #Forward | #Backward | Elapsed Time (sec) |
|---|---|---|---|---|---|
| LoRA | 64.3 | 1 | $B$ | $B$ | 39.7 |
| IncLoRA | 69.4 | $I$ | $B \times I$ | $B$ | 45.7 |
| O-IncLoRA | 76.6 | $I$ | $B \times I$ | $B + I \times r^2$ | 101.6 |
| SRB | 77.9 | 2 | $B$ | $B$ | 42.2 |

adapters. This orthogonality constraint adds computational complexity proportional to $I \times r^2$ during the backward pass (Wang et al., 2023), where $I$ is the number of tasks and $r$ is the rank of the adapters. Consequently, O-IncLoRA experienced significantly longer training times, as reflected in the elapsed time. In contrast, SRB requires only one additional adapter for the implicit task, resulting in two adapters (including the current adapter), regardless of the number of tasks. Moreover, SRB does not necessitate extra forward or backward passes through the network for each adapter. Instead, it performs simple arithmetic operations on the adapters, such as vector addition and scalar multiplication, which incur minimal computational overhead. As shown in Table 4, SRB achieved higher average accuracy than other methods while maintaining comparable elapsed time to LoRA.

## 6 CONCLUSION

In this paper, we addressed the limitations of current state-of-the-art CL methods for foundation LMs. These methods require task IDs that are difficult to obtain in real-world scenarios and often overlook recency bias. Focusing on task-agnostic settings, we introduced an implicit task to store historical knowledge while reducing recency bias where task IDs are not provided. By leveraging the implicit task as support for regularization, the proposed SRB maintains a balance between adapting to new tasks and retaining information from previous ones during the adapter's optimization process. As a result, SRB achieved superior performance compared to state-of-the-art methods with minimal additional computational overhead. These improvements are attributed to the SRB mechanism, which effectively retains past information by suppressing the recency bias that existing methods have overlooked. One limitation of our approach is the requirement for hyperparameters. However, SRB demonstrated consistent performance enhancements using fixed hyperparameters across task series of various orders, lengths, and different models. For future work, we plan to focus on implicitly identifying these hyperparameters, further enhancing the applicability and robustness of our method.

## REPRODUCIBILITY STATEMENT

In this paper, we conducted experiments based on the official CL benchmark as mentioned in Section 4. We also described more experimental details in Appendix C. We plan to make our code publicly available.

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

## A   Details of Suppressing Recency Bias

### A.1   Relationship between Adapter and Task Vector

From the perspective of task vectors, an adapter initializes to $\mathbf{0}$ and adapts to any task aligns with the definition of a task vector. Suppose the adapter $w_0$ is included in the initial weights $W_0$ and is fixed at zero. This adapter generates zero outputs during pretraining and does not participate in learning, making the process equivalent to pretraining without $w_0$. At time step $t$ of continual adaptation, the adapter becomes $w_t$. Given the zero initialization, the task vector $\tau_t$ is defined as $\tau_t = w_t - w_0 = w_t$. Therefore, a LoRA adapter with zero initialization can also be considered a task vector, and arithmetic operations can be applied to the adapter (Zhang et al., 2023b).

### A.2   Overall Process

---

**Algorithm 1** Suppressing Recency Bias

---

INPUT:

    Time step $t \in [1, T]$, Input data stream $(\mathcal{D}_1 \dots \mathcal{D}_T)$, Foundation LM weights $W_0$,

    Adapter weights $\{w_0, u_0\}$, Hyperparameters $(a, b, c)$,

    Initialization $w_0 \leftarrow \mathbf{0}$, $u_0 \leftarrow w_0$, $\tilde{w}_0 \leftarrow w_0$

**for** $t = 1, \dots, T$ **do**

    Optimization Process:

        $w_t \leftarrow \arg\max_{\tilde{w}_{t-1}} L(\{\tilde{w}_{t-1}, W_0\}, \mathcal{D}_t)$            ▷ Eq. (7)

    Arithmetic:

        // Current Task Vector:

        $\tau_t = w_t - w_0$

        // Implicit Task Vector:

        $\tau_t^u = a\tau_{t-1}^u + b(\tau_t - a\tau_{t-1}^u)$             ▷ Eq. (11)

        // Regularization Step:

        $\mathrm{oproj}_{\tau_t^u}(\tau_t) = \tau_t - \frac{\tau_t \cdot \tau_t^u}{\tau_t^u \cdot \tau_t^u}\tau_t^u$       ▷ Eq. (12)

        $\tilde{\tau}_t \leftarrow \tau_t - c \cdot \mathrm{oproj}_{\tau_t^u}(\tau_t)$           ▷ Eq. (13)

        $\tilde{w}_t \leftarrow \tilde{\tau}_t$

**end for**

---

Algorithm 1 presents the overall process of SRB. SRB operates over a total time step $T$ in task-agnostic continual learning settings, where tasks are not specified. The foundation LM weights $W_0$ remains frozen with only the fixed-size adapters $w_0$ and $u_0$ being updated. SRB is composed of two key processes: the first involves optimizing the adapters to encode knowledge from the current task, while the second performs arithmetic operations between the current task vector and the implicit task vector. This process preserves historical information and mitigates recency bias after the new task information is integrated through optimization.

## B   Related Works: Prompt and Optimization-based approach

Beyond replay-based methods that store information about past tasks and mix it with new tasks, recent studies in CL for foundation LMs can be broadly categorized into MAE and optimization-based. MAE stores task-specific information in separate modules, such as adapters or prompts, effectively leveraging previous knowledge. By combining the outputs of past and current modules, these methods preserve prior knowledge while adapting to new tasks. Among MAE methods, prompt-based methods optimize prompts, which are learnable embedding vectors rather than adapters. For instance, LFPT5 (Qin & Joty, 2021) learns soft prompts sequentially while generating task samples for replay. Similarly, the Progressive Prompt (ProgPrompt) (Razdaibiedina et al., 2023) adapts separate prompts for incoming downstream tasks and concatenates them sequentially with the previous prompts. Both LFPT5 and ProgPrompt mitigate catastrophic forgetting and adapt effectively to new tasks. However, they encounter challenges, including memory overhead caused by the extension of the soft prompt and the need for task IDs.

Optimization-based CL methods aim to limit changes to parameters that are important for retaining previous knowledge, often without expanding the model architecture. One such method, MagnItude-based Gradient Updating (MIGU) (Du et al., 2024) can be applied in a task-agnostic setting, unlike methods that require task IDs (Kirkpatrick et al., 2017b; Li & Hoiem, 2017). MIGU caches output magnitudes and updates only the parameters corresponding to the most significant values of the L1-normalized magnitudes. By leveraging the model's inherent features, MIGU effectively mitigates gradient conflicts and demonstrates stable performance in task-agnostic scenarios.

In addition to these foundational methods, O-IncLoRA (Wang et al., 2023) mitigates catastrophic forgetting by extending LoRA with task-specific orthogonal projections, preserving prior knowledge by minimizing task interference. However, it relies on explicit task IDs, limiting its effectiveness in task-agnostic settings, and independently applies orthogonal constraints for each task, leading to increased memory costs as tasks grow. In contrast, SRB eliminates the need for task IDs, dynamically integrates knowledge through implicit task vectors, and maintains a fixed computational footprint, enabling more efficient and scalable continual learning.

Besides, recent research has focused on understanding the dynamics of learning and forgetting during language model fine-tuning. (Zhang & Wu, 2024) investigates how fine-tuning affects different aspects of a language model's knowledge. The authors analyze the impact on elements such as topic, style, and factual knowledge, providing an in-depth examination of how fine-tuning can lead to biases or shifts in the model's behavior. By isolating these components, the study offers valuable insights into the internal mechanisms of language models, contributing to a better understanding of catastrophic forgetting and knowledge retention in continual learning scenarios.

## C    ADDITIONAL EXPERIMENTS DETAILS

### C.1    ORDERS OF TASK SERIES

Table 5: Six different task sequences used in continual learning experiments for checking forward transferability and generalization performance. The tasks correspond to the standard CL benchmarks adopted in previous studies.

| Order | Task Sequence |
| --- | --- |
| 1 | DBpedia → Amazon → Yahoo → AG News |
| 2 | DBpedia → Amazon → AG News → Yahoo |
| 3 | Yahoo → Amazon → AG News → DBpedia |
| 4 | MNLI → CB → WiC → COPA → QQP → BoolQ → RTE → IMDB → Yelp →Amazon → SST2 → DBpedia, → AG News → MultiRC → Yahoo |
| 5 | MultiRC → BoolQ → WiC → MNLI → CB → COPA → QQP → RTE → IMDB →SST2 → DBpedia → AG News → Yelp → Amazon → Yahoo |
| 6 | Yelp → Amazon → MNLI → CB → COPA → QQP → RTE → IMDB → SST2 → DBpedia → AG News → Yahoo → MultiRC → BoolQ → WiC |

### C.2    EXPERIMENT DETAILS

In this section, we provide specific experimental settings for each method. Our experiments were conducted using four NVIDIA GeForce RTX 3090 GPUs for T5 and four NVIDIA A100 for the LLaMA3 models.

**LoRA and IncLoRA**

- The batch size is set to 64.
- AdamW optimizer is used with hyperparameters $\beta_1 = 0.9$ and $\beta_2 = 0.999$.
- LoRA configuration: r = 8, $\alpha$ = 32, dropout = 0.05.
- The learning rate is set to 0.001 for the T5 model and 0.0001 for LLaMA3.

**O-IncLoRA**

- The threshold for mask selection is set at 0.7 across orders 1 to 6.
- All remaining hyperparameters are the same as those used in LoRA and IncLoRA.

**L2 regularization**

- The regularization rate $\lambda$ is set to 0.01.
- Training hyperparameters are consistent with LoRA and IncLoRA.

**MIGU**

- LoRA configuration: r = 8, $\alpha$ = 32, dropout = 0.05.
- The learning rate is set to 0.001.
- The threshold for mask selection is set at 0.7 across orders 1 to 6.

**ProgPrompt**

- The learning rate is set to 0.3.
- Prompt length: 50
- Task specific MLP layer is set as True.

**SRB**

- The hyperparameters (a,b,c) is set to (0.99, 0.025, 0.15) across all experiments.
- All remaining hyperparameters are consistent with those used in LoRA and IncLoRA.

## D  FURTHER EXPERIMENTAL RESULTS

Table 6: Average accuracy (Avg., %) on the CL benchmark for the T5 model, comparing results by method.

| Method | Order | | | Avg. | Order | | | Avg. | Task IDs |
|---|---|---|---|---|---|---|---|---|---|
| | 1 | 2 | 3 | | 4 | 5 | 6 | | |
| ProgPrompt | 75.2 | 75.0 | 75.1 | 75.1 | - | - | - | - | ✓ |
| LFPT5 | 77.1 | 76.2 | 76.6 | 76.6 | 68.4 | 68.8 | 71.4 | 69.5 | |
| LoRA | 60.6 | 62.1 | 68.8 | 63.8 | 53.7 | 44.4 | 39.8 | 46.0 | |
| MIGU | 74.8 | 71.6 | 73.5 | 73.3 | 66.9 | 64.8 | 51.8 | 61.2 | - |
| SRB | **78.1** | **78.2** | **77.5** | **77.9** | **70.5** | **71.4** | **73.3** | **71.7** | |

Table 6 presents the average accuracy of prompt-based CL methods and task-agnostic CL methods on the standard CL and long CL benchmarks. While MIGU and LoRA can be applied without task IDs, they performed poorly on both benchmarks compared to prompt-based CL methods such as LFPT5 and ProgPrompt, which require task IDs. These results show that task-specific information has a significant impact on CL performance. Notably, SRB does not require task IDs such as MIGU and LoRA, yet it achieved performance improvements of approximately 1.3% and 2.2% over LFPT5 on the standard and long CL benchmarks, respectively. It demonstrates that SRB effectively leverages past knowledge and adapts to new tasks, making it applicable to general CL environments with or without task IDs.

## E  FURTHER DISCUSSIONS

### E.1  EFFECTIVENESS OF RECOVERY ARITHMETIC

Table 7 examines the effect of the recovery operation in Eq. (11), which preserves the foundation LM weights more during adaptation, on performance across different models. For the T5 model on the standard CL benchmark, we observed that not performing the recovery led to a slight increase

Table 7: Average accuracy (Avg., %) on the CL benchmark, comparing results with and without Recovery applied. When Recovery is applied, $a = 0.99$; when it is not applied, $a = 1.0$.

| Model | Recovery | Order 1 | Order 2 | Order 3 | Avg. |
|---|---|---|---|---|---|
| | ✓ | 78.1 | 78.2 | **77.5** | 77.9 |
| | - | **78.5** | **78.8** | 77.1 | **78.1** |
| T5 | Recovery | Order 4 | Order 5 | Order 6 | Avg. |
| | ✓ | 70.5 | **71.4** | **73.3** | **71.7** |
| | - | **70.9** | 67.8 | 71.0 | 69.9 |
| Model | Recovery | Order 1 | Order 2 | Order 3 | Avg. |
| LLaMA3 | ✓ | **79.0** | **80.5** | **77.0** | **78.8** |
| | - | 78.8 | 79.7 | 76.8 | 78.5 |
| LLaMA3-chat | ✓ | **78.9** | 80.3 | **78.0** | **79.1** |
| | - | 78.0 | **81.1** | 77.6 | 78.9 |

in performance of approximately 0.2%. However, in the case of longer task sequences, performing the recovery resulted in a performance improvement of 1.8%. This trend is consistent for the LLaMA models. The recovery strategy increased performance by 0.3% for LLaMA3 and by 0.2% for LLaMA3-chat. These results suggest that the recovery strategy is more effective when dealing with a more significant number of tasks or starting from a model with higher initial performance.

### E.2 EFFECTIVENESS OF UPDATE ARITHMETIC

Table 8: Average accuracy (%) of the standard CL benchmark for T5 as $b$ vary.

| Hyperparameter | Method SRB | | | | | | LoRA |
|---|---|---|---|---|---|---|---|
| | 0.015 | 0.02 | 0.025 | 0.03 | 0.05 | 0.1 | 75.8 |
| $b$ | 77.4 | 77.5 | 77.9 | **78.3** | 77.6 | 75.7 | |

Table 8 presents the performance changes of our SRB method as we vary the hyperparameter $b$ in Eq. (11). The hyperparameter $b$ controls the extent to which information with reduced recency bias is updated for new tasks. Our experimental results show that SRB performed well when $b$ is less than 0.1, with a tendency for performance to decrease when $b$ exceeds 0.03. This indicates that updating new task information relatively slowly—thereby strengthening the low-pass filter characteristic—is crucial for the performance.

### E.3 EXPERIMENTAL RESULTS OF APPLYING LoRA TO VARYING ATTENTION WEIGHTS

Table 9: Performance comparison of LoRA, IncLoRA, and SRB applied to query (q), value (v), and both q and v across standard CL experiments (Order 1, 2, and 3) and long CL experiments (Order 4, 5, and 6).

| Method | Target | Order 1 | Order 2 | Order 3 | Avg. | Order 4 | Order 5 | Order 6 | Avg. |
|---|---|---|---|---|---|---|---|---|---|
| | q,v | 61.9 | 62.1 | 68.8 | 64.3 | 53.7 | 44.4 | 39.8 | 46.0 |
| LoRA | q | 72.5 | 70.8 | 67.6 | 70.3 | 57.4 | 60.2 | 34.1 | 50.6 |
| | v | 70.2 | 68.6 | 71.3 | 70.0 | 66.2 | 58.0 | 13.3 | 45.9 |
| | q,v | 71.4 | 66.2 | 70.7 | 69.4 | 62.3 | 66.2 | 63.5 | 64.0 |
| IncLoRA | q | 76.2 | 75.2 | 74.5 | 75.3 | 64.1 | 65.1 | 67.0 | 65.4 |
| | v | 73.9 | 67.9 | 68.4 | 70.0 | 66.1 | 63.3 | 60.9 | 63.4 |
| | q,v | 78.1 | **78.2** | **77.5** | 77.9 | 70.5 | 71.4 | 73.3 | 71.7 |
| SRB | q | **78.3** | 78.1 | 77.4 | **78.0** | 67.4 | 69.3 | 69.0 | 68.6 |
| | v | 72.8 | 68.5 | 71.9 | 71.1 | 63.8 | 64.1 | 68.3 | 65.4 |

We measured the performance variations when applying LoRA to different attention weights (query, value, and both query and value) across the standard and the long CL benchmark in Table 9. Our

experimental results demonstrated that SRB consistently achieved the highest average accuracy, regardless of the attention weights used.

