# OpenReview forum: "Suppressing recency bias through implicit task in task-agnostic continual adaptation for foundation language models"
_ICLR.cc/2025/Conference — ICLR 2025 Conference Withdrawn Submission_

### Official Review · Reviewer_MLub · 2024-11-02

**Soundness:** 2
**Presentation:** 3
**Contribution:** 2
**Rating:** 3
**Confidence:** 3

**Summary:**

This paper addresses the problem of recency bias in foundation models, which causes models to prioritize recent tasks at the expense of past knowledge. The main contribution of this work is a novel method called "Suppressing Recency Bias"(SRB), which combines current and past knowledge using arithmetic operations without requiring additional back-propagation, thus ensuring minimal computational overhead. The key highlights of SRB that authors claim include:
1. SRB eliminates the need for task IDs by using an implicit task that aggregates past knowledge through simple arithmetic.
2. SRB maintains historical knowledge from past tasks while learning new tasks by integrating only unique information, thus reducing redundant learning.
3. SRB requires minimal additional memory space and computations to SOTA methods, outperforming methods like LoRA and IncLoRA effectively.

**Strengths:**

1. It is a novel approach to handling one of the major challenges in continual learning: recency bias. By leveraging an implicit task to integrate historical knowledge, SRB addresses the common problem where models tend to overly focus on recent tasks at the expense of retaining knowledge from earlier ones. This approach could be beneficial across a wide range of continual learning applications.

2. SRB operates without requiring task IDs, which is a significant advantage in real-world scenarios where tasks are not explicitly defined. Task-agnostic continual learning is particularly challenging, and the SRB method makes a good improvement by showing it’s possible to achieve effective continual adaptation without relying on task-specific information.

3. The experimental results show SRB outperforms or is competitive with state-of-the-art methods like LoRA, IncLoRA, and O-IncLoRA on some standard CL benchmarks. SRB demonstrates superior generalization and knowledge retention across tasks, particularly in comparison to methods prone to catastrophic forgetting.

4. The authors provide a clear description of the experimental setup, benchmark datasets, and hyperparameters settings. This transparency supports reproducibility of this work.

**Weaknesses:**

The paper presents a novel approach to addressing recency bias in task-agnostic continual learning. However, there are some potential weaknesses and limitations in this work.

1. The SRB method introduces several hyperparameters (such as a, b, and c for controlling the influence and regularization of task vectors). The paper notes that hyperparameter tuning is essential for SRB’s performance. This reliance might limit the model’s robustness, as it may require fine-tuning for different tasks or models, reducing its practicality in real-world, dynamic settings where such tuning isn’t feasible.

2. Although SRB is designed for task-agnostic settings, the scalability to larger or longer task sequences is not thoroughly explored. For instance, the implicit task mechanism might become less efficient or struggle to represent historical knowledge accurately when handling an extensive range of tasks. A larger-scale experiment would provide insights into how SRB performs with extensive, varied task sequences.

3. Since SRB relies on arithmetic operations to balance historical and current knowledge, it may struggle in environments where task characteristics change quickly or drastically. The implicit task representation could fail to adapt promptly in such settings, potentially limiting SRB’s performance on tasks that require quick, context-sensitive adaptation.

**Questions:**

1.  In the regularization process (Equation 13), how to determine the effectiveness of the orthogonal projection for suppressing recency bias? Were there other regularization techniques or projections you considered, and if so, what made this one preferable? In addition, if the implicit task vector is orthogonal to \tau_t, then the projections becomes zero, is there any additional strategy to prevent this situation?

2.  In task-agnostic settings, are there specific application domains where SRB’s approach to suppressing recency bias is particularly valuable? Conversely, are there domains where the implicit task approach may struggle, such as with tasks that have low overlap or are highly distinct?

3. You mention using average accuracy for performance evaluation, but did you also measure other indicators like backward and forward transfer? If so, how did SRB perform on these metrics, especially in preserving knowledge from earlier tasks?

---

> ### Author Response · Authors · 2024-11-21
> **Response to reviewer**
>
> **1. The SRB method introduces several hyperparameters (such as a, b, and c for controlling the influence and regularization of task vectors). The paper notes that hyperparameter tuning is essential for SRB’s performance. This reliance might limit the model’s robustness, as it may require fine-tuning for different tasks or models, reducing its practicality in real-world, dynamic settings where such tuning isn’t feasible.**
>
> **A1.** Thank you for pointing out the dependency on hyperparameters. Reducing the reliance on manual tuning is important, especially in dynamic real-world environments. **However, under fixed hyperparameters, SRB consistently outperforms the SOTA method on a variety of benchmarks and models without the extensive optimizations in Section 4.3.** As mentioned in Section 6, exploring adaptive or meta-learned hyperparameter strategies is a promising direction for future work to improve the practicality and robustness of SRB.
>
> **2. Although SRB is designed for task-agnostic settings, the scalability to larger or longer task sequences is not thoroughly explored. For instance, the implicit task mechanism might become less efficient or struggle to represent historical knowledge accurately when handling an extensive range of tasks. A larger-scale experiment would provide insights into how SRB performs with extensive, varied task sequences.**
>
> **A2.** While our experiments included long sequences with up to 15 tasks (shown in Section 4.2), testing SRB on much larger and more diverse task sequences would provide deeper insights into scalability. **However, as mentioned in Section 4.1, the benchmarks we used in our study followed standard scenarios that have been extensively studied previously for LM and LLM.**
>
> **3. Since SRB relies on arithmetic operations to balance historical and current knowledge, it may struggle in environments where task characteristics change quickly or drastically. The implicit task representation could fail to adapt promptly in such settings, potentially limiting SRB’s performance on tasks that require quick, context-sensitive adaptation.**
>
> **A3.** We appreciate your suggestion of extended experiments to validate our work. For such validation, it is worthwhile to study improving the existing standard benchmarks (Section 4.1). Nevertheless, we believe that SRB, which shows robust performance in a variety of environments, is designed to allow regularization strategies that reflect the distance between the implicit task and the current task to perform effectively in rapidly changing experimental scenarios (Section 3.4).

---

> ### Author Response · Authors · 2024-11-21
> **Continued**
>
> **Q1. In the regularization process (Equation 13), how to determine the effectiveness of the orthogonal projection for suppressing recency bias? Were there other regularization techniques or projections you considered, and if so, what made this one preferable? In addition, if the implicit task vector is orthogonal to $\tau_t$, then the projections becomes zero, is there any additional strategy to prevent this situation?**
>
> **Q1-A.** Thank you for raising this insightful question. The orthogonal projection suppresses recency bias by dynamically balancing the influence of historical and current task vectors. Its effectiveness was validated through experiments showing SRB's ability to retain performance on earlier tasks (Section 5.1). While we considered other regularization techniques, including L2-based penalties, we found the projection method preferable due to its ability to explicitly measure and adjust for alignment between task vectors.
>
> In addition, hyperparameter $c$ in our approach is specifically designed to balance past and current information during the regularization process (Section 3.4). When the implicit task vector becomes orthogonal to $\tau_t$, it indicates that past knowledge and the current task information do not interfere with each other. This scenario suggests that the information captured by these task vectors is independent, which aligns with the goal of preserving historical knowledge without redundancy.
>
> **Q2. In task-agnostic settings, are there specific application domains where SRB’s approach to suppressing recency bias is particularly valuable? Conversely, are there domains where the implicit task approach may struggle, such as with tasks that have low overlap or are highly distinct?**
>
> **Q2-A.** We appreciate your thoughtful question regarding SRB’s applicability in task-agnostic settings. When models are exposed to data from the same context or domain repeatedly, there is a risk of overfitting to the most recent tasks, which can diminish the foundation model's generalization capabilities. By mitigating recency bias, SRB ensures that the foundational abilities of the model are preserved, allowing it to perform robustly across diverse tasks and domains.
> For domains with low overlap or highly distinct tasks, the implicit task vector mechanism effectively addresses this challenge. As observed in domain transfer scenarios, the second term of Eq. (11) limits significant updates to the implicit task vector when there is little overlap between the tasks. This helps SRB maintain stability and prevents unnecessary adjustments, ensuring that the model adapts appropriately to new tasks without eroding previously learned knowledge.
>
> **Q3. You mention using average accuracy for performance evaluation, but did you also measure other indicators like backward and forward transfer? If so, how did SRB perform on these metrics, especially in preserving knowledge from earlier tasks?**
>
> **Q3-A.** While average accuracy is a primary evaluation metric, we also examined forward transfer and preservation of knowledge from earlier tasks, as discussed in Section 5.1. SRB demonstrated strong performance in forward transfer by leveraging implicit task vectors to retain generalizable features. Similarly, backward transfer showed that SRB effectively mitigates forgetting by suppressing recency bias, as evident in the nearly parallel performance trends over time (Figures 3(b)–3(d)). These results confirm SRB's capacity to balance historical knowledge retention with new task adaptation.

---

### Official Review · Reviewer_9Jp6 · 2024-11-02

**Soundness:** 2
**Presentation:** 3
**Contribution:** 3
**Rating:** 6
**Confidence:** 2

**Summary:**

This paper investigates task-agnostic continual learning using foundational language models. To tackle the issues associated with previous PEFT methods, the authors introduce a novel approach called Suppressing Recency Bias (SRB), which enables the model to adapt to current data while retaining historical knowledge. By leveraging the design of implicit task vectors, SRB-based models can be trained to adapt without the need for task IDs. Experimental results demonstrate the effectiveness of the proposed method.

**Strengths:**

1. The idea of employing a fixed-size adapter to recursively store the historical knowledge of an implicit task appears to be interesting and novel.  The proposed method can adapt to task-agnostic datasets even without task IDs.
2. The authors have conducted a thorough series of experiments to validate their proposed method.
3. The results regarding information retention indicate that the SRB significantly outperforms other models, showcasing its ability to adapt to new tasks while preserving performance on previous ones with minimal degradation.

**Weaknesses:**

1. The paper lacks a detailed explanation of the motivation behind the model design, particularly concerning the implicit task vector and the regularization term.
2. The authors should apply the proposed method to additional foundational models (e.g., BERT) to further validate its effectiveness, similar to previous studies.
3. The explanation of how to calculate the task vector is somewhat unclear. For instance, as stated in line 288, the task vector for the current task is defined as $\tau_t=w_t-w_0$. However, the process for calculating the task vector for the next task is not elaborated upon.

**Questions:**

1. Why does the implicit task vector act as a low-pass filter, limiting diversity?
2. What are the results of LLaMA3 and LLaMA3-chat on Orders 4, 5, and 6?

---

> ### Author Response · Authors · 2024-11-21
> **Response to reviewer**
>
> **1. The paper lacks a detailed explanation of the motivation behind the model design, particularly concerning the implicit task vector and the regularization term.**
>
> **A1.** We appreciate you pointing out this area for clarification. The motivation behind SRB’s design, particularly the implicit task vector and the regularization term, is to address **recency bias** and **catastrophic forgetting** in task-agnostic continual learning. As mentioned in Section 3.3, the value of $b$ plays a crucial role: when set close to 0, it causes the model to respond slowly to rapidly changing information, effectively acting as a **low-pass filter**. This ensures that SRB prioritizes stable, long-term knowledge over short-term fluctuations, mitigating the risk of overfitting to recent tasks.
>
> The implicit task vector $(τ_u)$ aggregates historical knowledge, and allows SRB to retain relevant information from past tasks while minimizing interference from noisy or task-specific updates. The regularization term Eq. (13) complements this filtering mechanism by constraining parameter updates orthogonally to the implicit task vector. This dynamic adjustment ensures a balance between retaining historical knowledge and adapting to new tasks.
>
> **2. The authors should apply the proposed method to additional foundational models (e.g., BERT) to further validate its effectiveness, similar to previous studies.**
>
> **A2.** Thank you for this thoughtful suggestion. Evaluating SRB on BERT or other foundational models is indeed a valuable direction for further validation. In our current work, we focused on two large language models (LLMs) from the LLaMA family to demonstrate SRB’s robustness and effectiveness across diverse and challenging task-agnostic continual learning scenarios. This choice was motivated by the increasing adoption of LLMs in real-world applications, making them highly relevant for this study.
> Our results showed consistent performance improvements across these LLMs, as detailed in Tables 2 and 3, highlighting SRB’s generalizability.
>
> **3. The explanation of how to calculate the task vector is somewhat unclear. For instance, as stated in line 288, the task vector for the current task is defined as $τ_t=w_t−w_0$. However, the process for calculating the task vector for the next task is not elaborated upon.**
>
> **A3.** The task vector $\tau_t$ is calculated as the difference between the current task’s weight vector $w_t$ and the initial weight vector $w_0$ of the foundation model. When transitioning to the next task $t+1$, the task vector is updated incrementally to capture the cumulative effect of historical and new task knowledge. In line 332-334, this incremental update occurs as:
>
> $\tau_{t+1} = w_{t+1} - w_0$.
>
> **Q1. Why does the implicit task vector act as a low-pass filter, limiting diversity?**
>
> **Q1-A.** Thank you for this insightful question. As mentioned in **Section 3.3** of the paper, the implicit task vector $\tau_u$ acts as a low-pass filter because it is designed to prioritize stable, long-term information over rapidly changing, task-specific updates. This behavior is primarily influenced by the hyperparameter $b$, which, when set close to 0, ensures that the model responds slowly to new information, suppressing noisy or transient updates.
>
> This low-pass filtering property is intentional and helps mitigate recency bias, a common issue in task-agnostic continual learning, where the model might overly adapt to recent tasks at the expense of earlier ones. While this mechanism may limit diversity, it ensures robust retention of essential historical knowledge, as demonstrated in Tables 2 and 3, where SRB outperforms other baselines in maintaining performance across diverse task sequences.

---

> > ### Comment · Reviewer_9Jp6 · 2024-11-26
> > **Keep rating at 6**
> >
> > I've read the authors' rebuttal and other reviewers' comments, especially those who gave low ratings. In general, the method seems incremental and the comparison is not complete (pointed out by jTKg, and the authors admitted it). Given all of these, I choose to keep the rating of 6, but I will lower my confidence further.

---

> > > ### Author Response · Authors · 2024-11-27
> > > **Response to reviewer 9Jp6**
> > >
> > > Thank you for your valuable and insightful review. Our work specifically addresses **task-agnostic continual learning**, expanding beyond the traditional perspective of incremental learning. Research in this area has been relatively sparse, which is why we conducted a comparison with the state-of-the-art method, **O-IncLoRA**, to highlight SRB’s performance.
> > >
> > > While we agree that applying a broader range of foundational continual learning methods could diversify research in the field of **continual learning for LMs**, we respectfully disagree with the notion that our comparisons lack completeness. Our study was carefully designed to provide a meaningful and robust evaluation of SRB within the task-agnostic continual learning context.

---

### Official Review · Reviewer_CMCi · 2024-11-02

**Soundness:** 3
**Presentation:** 3
**Contribution:** 3
**Rating:** 3
**Confidence:** 3

**Summary:**

The paper presents **Suppressing Recency Bias (SRB)**, a method designed for task-agnostic continual learning in foundation language models (LMs). SRB introduces the concept of an implicit task that integrates knowledge recursively, minimizing the reliance on task identifiers and addressing recency bias—an issue where models disproportionately prioritize current tasks at the expense of previous ones. This approach achieves low memory overhead by requiring only fixed-size adapters and using simple arithmetic operations for updating, without the need for backpropagation. The paper demonstrates that SRB outperforms existing continual learning (CL) methods in both standard and extended task sequences by maintaining superior performance across diverse tasks and reducing recency bias.

**Strengths:**

- **Task-Agnostic Adaptability**: SRB excels in task-agnostic settings, removing the dependency on task identifiers and preserving performance across tasks.
- **Computational Efficiency**: The method achieves this while maintaining minimal computational and memory overhead, only needing simple arithmetic operations.
- **Reduction of Recency Bias**: The introduction of implicit task vectors effectively mitigates recency bias, ensuring that historical information is preserved during adaptation.
- **Empirical Validation**: Experimental results on standard and long CL benchmarks show that SRB outperforms other state-of-the-art methods in accuracy and efficiency.

**Weaknesses:**

- **Hyperparameter Sensitivity**: The approach relies on specific hyperparameters for performance, which might affect its generalizability without tuning.
- **Comparison Scope**: While the paper benchmarks against key methods, additional comparisons with more diverse baseline approaches, such as advanced replay-based strategies, could strengthen its conclusions.
- **Task Transition Analysis**: The paper could benefit from deeper analysis of how SRB handles transitions between tasks, especially in complex sequences involving highly dissimilar tasks.
- **Overhead of Hyperparameter Tuning**: Although SRB shows robustness, the paper notes fixed hyperparameters, implying that different task sequences might require adjustment for optimal performance.

**Questions:**

- Can the authors elaborate on how the method performs under highly diverse task sequences, such as tasks involving distinct domains (e.g., code, legal text, medical literature)?
- What strategies can be used to fine-tune the hyperparameters (a, b, c) without extensive trial and error?
- How does SRB handle tasks that might involve conflicting objectives (e.g., creative writing vs. technical report summarization)?

---

> ### Author Response · Authors · 2024-11-21
> **Response to reviewer**
>
> **Q1. Hyperparameter Sensitivity: The approach relies on specific hyperparameters for performance, which might affect its generalizability without tuning.**
>
> **A1.** Thank you for pointing out the potential concern regarding hyperparameter sensitivity. While our approach relies on specific hyperparameters $(a, b, c)$, we have demonstrated Appendix E that these parameters exhibit robust performance across diverse task sequences without extensive tuning. **Moreover, under fixed hyperparameters, SRB consistently outperforms the SOTA method on a variety of benchmarks and models without the extensive optimizations in Section 4.3.**  As mentioned in Section 6, exploring adaptive or meta-learned hyperparameter strategies is a promising direction for future work to improve the practicality and robustness of SRB.
>
> **Q2. Comparison Scope: While the paper benchmarks against key methods, additional comparisons with more diverse baseline approaches, such as advanced replay-based strategies, could strengthen its conclusions.**
>
> **A2.** Thank you for highlighting the importance of broader comparisons. Our primary focus was to demonstrate the efficacy of SRB in **task-agnostic continual learning** settings. Unlike traditional methods, SRB does not rely on task identifiers and instead employs **simple vector arithmetic** to suppress recency bias and preserve historical knowledge effectively.  To emphasize SRB’s strengths in task-agnostic scenarios, we selected **IncLoRA** and **O-IncLoRA** as key baselines, as these methods represent state-of-the-art performance in parameter-efficient fine-tuning and continual learning.
>
> **Q3. Task Transition Analysis: The paper could benefit from deeper analysis of how SRB handles transitions between tasks, especially in complex sequences involving highly dissimilar tasks.**
>
> **A3.** Thank you for highlighting this point. A deeper analysis of task transitions, particularly across highly dissimilar tasks, would provide valuable insights. Section 5.1 discusses the ability of SRB to mitigate recency bias and preserve historical knowledge during task transitions.

---

> ### Author Response · Authors · 2024-11-21
> **Continued**
>
> **Q5. Can the authors elaborate on how the method performs under highly diverse task sequences, such as tasks involving distinct domains (e.g., code, legal text, medical literature)?**
>
> **A5.** Our experiments (e.g., Section 4.2 and Table 2) demonstrate SRB's robustness across benchmarks involving diverse datasets such as AG News, DBpedia, and tasks from GLUE and SuperGLUE. While these tasks vary in domain and complexity,  incorporating even more diverse domains (e.g., code, legal text, medical literature) could further validate SRB’s adaptability. This is an area we plan to explore in future work by expanding the benchmark to include tasks with broader domain-specific challenges.
>
> **Q6. Overhead of Hyperparameter Tuning: Although SRB shows robustness, the paper notes fixed hyperparameters, implying that different task sequences might require adjustment for optimal performance.Q6. What strategies can be used to fine-tune the hyperparameters $(a, b, c)$ without extensive trial and error?**
>
> **A6.** Thank you for this insightful question. To fine-tune the hyperparameters $(a, b, c)$without extensive trial and error, we recommend the following approach:
>
> The hyperparameter $a$ plays a critical role in controlling the balance between historical knowledge and current information. Since SRB aims to suppress recency bias effectively, $a$ should be set close to 1 to emphasize the **low-pass filter** property. This ensures that the implicit task vector retains a significant portion of historical knowledge, preventing the model from overly adapting to recent tasks at the expense of prior information.
>
> In our experiments, we observed that $a=0.99$ consistently performed well across various task sequences (Section 4.3). This suggests that fine-tuning $a$ does not require extensive adjustments, as its optimal range is relatively stable. Similarly, the parameters $b$ and $c$ can be adjusted within smaller ranges (e.g., 0.01–0.1) based on the task sequence dynamics.
>
> Despite the need for some hyperparameter tuning, as shown in Sections 4.2 and 4.3, SRB demonstrated **consistent performance improvements** across multiple models and diverse dataset scenarios. This consistency highlights SRB’s robustness, as also noted in the conclusion. We will include this guidance in the revised paper to help practitioners adopt SRB with minimal tuning efforts.
>
> **Q7. How does SRB handle tasks that might involve conflicting objectives (e.g., creative writing vs. technical report summarization)?**
>
> **A7.**  Thank you for this thoughtful question. As shown in Table 2, SRB demonstrated consistent performance improvements regardless of the task order, indicating its robustness in handling a variety of task sequences. However, the observed performance differences across orders highlight the challenge posed by tasks with conflicting objectives, such as creative writing and technical summarization. This reflects the impact of task characteristics and order on the implicit task representation, as you have pointed out.
>
> Addressing these differences is an important area for further exploration. Despite this, the results illustrate the potential of SRB in managing conflicting objectives within a task-agnostic continual learning framework. We plan to investigate such extreme scenarios more thoroughly in future work, focusing on refining task representations to better handle rapidly changing or highly distinct objectives.

---

### Official Review · Reviewer_X9bG · 2024-11-03

**Soundness:** 2
**Presentation:** 2
**Contribution:** 2
**Rating:** 5
**Confidence:** 3

**Summary:**

This paper propose a continual learning algorithm that projects LoRA parameter updates to a subspace defined by an "implicit task" to mitigate recency bias and mitigate forgetting. Experiments on CL benchmarks demonstrate performance improvements over baselines, especially those that also learn LoRA.

**Strengths:**

- The performance improvement is clear compared to the baselines.
- The description of the proposed approach is clear

**Weaknesses:**

1. The intuition behind the performance improvement is not quite clear to me.

In a high level, the approach finds a direction of model parameter update that represents previous tasks (termed as "implicit tasks" in this work), and apply regularization in weight updates while learning new tasks by "pushing back" the update a bit towards a direction orthogonal to the "implicit tasks".

- It seems Orthogonal LoRA by Wang et al. 2023 does a similar job. What is the design that makes the approach improve over Orthogonal LoRA?  Please highlight the key methodological differences and explain how these differences contribute to the improved performance observed in the experiments.

- The approach conceptually shares similar idea to regularization based CL approaches like L2 regularization (which pushes back parameter updates to their initial states before fine-tuning). But in Table 2 L2 regularization performs very poorly. What could be the reason? Please provide a more in-depth analysis of why the proposed approach outperforms L2 regularization.

**Questions:**

See weakness.

---

> ### Author Response · Authors · 2024-11-21
> **Response to reviewer**
>
> # Overall
> We sincerely appreciate reviewer X9bG's thoughtful questions and have carefully considered them to refine and clarify our work. In particular, we have added details to **Appendix B** to further elaborate on the differences between SRB and O-LoRA, ensuring a more comprehensive comparison that highlights the methodological distinctions and their implications for performance.
>
> # Response
> **Q1. The intuition behind the performance improvement is not quite clear to me. In a high level, the approach finds a direction of model parameter update that represents previous tasks (termed as "implicit tasks" in this work), and apply regularization in weight updates while learning new tasks by "pushing back" the update a bit towards a direction orthogonal to the "implicit tasks".**
>
> **A1.** Thank you for raising these insightful points. The intuition behind SRB’s performance improvement lies in its ability to dynamically preserve knowledge from previous tasks without requiring explicit task identifiers or boundaries. By leveraging implicit task vectors, SRB effectively captures the core representation of past tasks. During the learning process for a new task, SRB applies regularization through orthogonal projections, as described in Section 3.4.
>
> This regularization ensures that updates to the model parameters are constrained in a way that:
>
> 1. Minimizes interference with previously learned knowledge by "pushing back" updates orthogonally to the implicit task vector.
> 2. Allows sufficient flexibility to adapt to the unique requirements of the new task.
>
> This mechanism balances the trade-off between retaining historical information and adapting to new tasks, enabling SRB to suppress recency bias and mitigate catastrophic forgetting, as demonstrated in Tables 2 and 3.
>
> **Q2. It seems Orthogonal LoRA by Wang et al. 2023 does a similar job. What is the design that makes the approach improve over Orthogonal LoRA? Please highlight the key methodological differences and explain how these differences contribute to the improved performance observed in the experiments.**
>
> **A2.** Thank you for raising this important question. While Orthogonal LoRA (O-LoRA) and SRB share the objective of preserving prior knowledge through orthogonal subspace learning, SRB demonstrates superior performance due to the following key design advantages:
>
> - Task-Agnostic Setting:
>     - O-LoRA requires explicit task identifiers for constructing task-specific orthogonal projections. This dependency makes O-LoRA less effective in task-agnostic settings where such identifiers are unavailable.
>     - SRB, in contrast, is designed for task-agnostic continual learning. It uses implicit task vectors dynamically constructed from historical information, eliminating the need for task IDs and enabling SRB to generalize more effectively across diverse and sequential tasks.
> - Dynamic Knowledge Integration:
>     - O-LoRA independently applies orthogonal constraints for each task, which can result in inefficiencies when tasks overlap or share commonalities.
>     - SRB leverages vector arithmetic to dynamically integrate knowledge from prior and current tasks. This ensures that updates reflect the nuanced relationships between tasks, enabling better adaptation to task sequences.
> - Efficient Regularization:
>     - O-LoRA’s task-specific orthogonal projections may inadvertently over-constrain updates, particularly for ambiguous or overlapping tasks.
>     - SRB’s lightweight regularization mechanism (Equation 11) selectively constrains updates based on the implicit task vector, preserving critical parameters for prior tasks while allowing flexibility for current task learning.
> - Reduced Computational Overhead:
>     - O-LoRA requires additional task-specific parameters for each task, which increases memory and computational demands as the number of tasks grows.
>     - SRB maintains a fixed computational footprint by relying on a single implicit task vector, making it scalable to longer task sequences.

---

> ### Author Response · Authors · 2024-11-21
> **Continued**
>
> **Q3. The approach conceptually shares similar idea to regularization based CL approaches like L2 regularization (which pushes back parameter updates to their initial states before fine-tuning). But in Table 2 L2 regularization performs very poorly. What could be the reason? Please provide a more in-depth analysis of why the proposed approach outperforms L2 regularization.**
>
> **A3.** The SRB approach and L2 regularization share a conceptual similarity in constraining parameter updates, but their mechanisms differ fundamentally, leading to the observed performance gap:
>
> 1. Selective Regularization:
>     - L2 regularization pushes all parameters back to their initial states, which can over-constrain updates and hinder adaptation to new tasks, especially when task characteristics differ significantly.
>     - SRB, in contrast, applies selective regularization by projecting updates orthogonally to the implicit task vector, ensuring that only parameters irrelevant to previous tasks are adjusted for the new task.
> 2. Task Representation:
>     - L2 regularization lacks a mechanism to explicitly represent past task knowledge, treating all updates uniformly.
>     - SRB leverages implicit task vectors to capture the core knowledge of prior tasks, allowing it to selectively retain relevant historical information.
> 3. Mitigation of Recency Bias:
>     - L2 regularization does not address recency bias directly, often leading to the overwriting of historical knowledge.
>     - SRB explicitly targets recency bias through its orthogonal projection mechanism, maintaining a balance between historical retention and current task adaptation.
>
> These differences explain why SRB significantly outperforms L2 regularization, as evidenced in Table 2, by preserving task-specific knowledge while maintaining flexibility for new tasks.

---

### Official Review · Reviewer_xDBY · 2024-11-03

**Soundness:** 3
**Presentation:** 3
**Contribution:** 3
**Rating:** 6
**Confidence:** 4

**Summary:**

This paper introduces the Suppressing Recency Bias (SRB) method to mitigate catastrophic forgetting during the continuous learning process of language models (LMs) without task IDs. SRB introduces an additional implicit task adapter and designs an update mechanism to appropriately integrate knowledge learned from the current task into the implicit adapter. The updated implicit adapter is then used to initialize the learning of new data. The designed update mechanism reduces duplicated information in classical Model Architecture Expansion (MAE) methods while balancing the increase in adapter diversity and the reduction of recency bias. The method outperforms previous task-agnostic methods and MAE methods using task IDs on benchmark tasks. Ablation studies demonstrate the effects of hyperparameters on reducing recency bias and increasing diversity, and show that SRB is not sensitive to hyperparameters within a certain range.

**Strengths:**

- The paper is clearly written and easy to understand. Figure 1 effectively contrasts SRB with other methods, and Figure 2 helps in understanding the purpose of the update mechanisms in SRB.
- SRB achieves a balance between increasing adapter diversity and reducing recency bias through a carefully designed update mechanism, which is validated by detailed ablation studies.
- SRB is simple and easy to use, and it outperforms existing MAE methods in terms of computational and memory costs.

**Weaknesses:**

- The paper lacks a clear explanation of the division and training situation for $(D_1, …, D_T)$ in the SRB setting. Does each $D_t$ correspond to the data contained in a default training batch, or is it a manually divided portion of the dataset?
  - If it corresponds to data in a single batch, how many times will this data be updated? What is the impact of different batch sizes on the relevant hyperparameters?
  - If it is a manually divided portion of the dataset, how is the division performed? If the division ensures that data from one task forms a single $D_i$, then SRB actually leaks task ID information. If data from each task is evenly divided into N parts, it still contains some task information. In a realistic task-agnostic scenario, it is difficult to ensure that data from different tasks are divided into different groups. The reviewer would like to see the performance of SRB when data from sequential tasks are included in the same $D_t$.
- The results section lacks methods such as multi-task learning or task experts as performance upper bounds for reference. Adding this reference would help readers better understand the improvements and limitations of the proposed method. Additionally, Figure 3(a) in Section 5.1 could also provide the performance of task experts as a reference.

**Questions:**

- Please provide more detailed explanations regarding the division of $D_i$ as mentioned in the weaknesses section.

---

> ### Author Response · Authors · 2024-11-21
> **Response to reviewer**
>
> **Q1. The paper lacks a clear explanation of the division and training situation for $(D1,…,DT)$  in the SRB setting. Does each $D_t$ correspond to the data contained in a default training batch, or is it a manually divided portion of the dataset?**
>
> **A1.** Thank you for raising this important question about how datasets $D_t$ are divided and trained within the SRB framework. In our work, $D_t$ corresponds to mini-batches sequentially sampled during training, as described in Section 3.1. These mini-batches are not manually divided but are naturally derived from the data stream, ensuring that SRB operates in a truly task-agnostic manner without leaking task ID information.
>
> **Q2. If it corresponds to data in a single batch, how many times will this data be updated? What is the impact of different batch sizes on the relevant hyperparameters?**
>
> **A2.** In our experiments, each batch is processed **exactly once** during training. To examine the impact of different batch sizes on SRB’s performance, we conducted additional experiments by varying the batch size while keeping all other settings constant. The results are summarized below:
>
> | Batch size | Order  1 |  Order 2 | Order3 | Avg. |
> | --- | --- | --- | --- | --- |
> | 8 | 77.0 | 77.8 | 77.0 | 77.3 |
> | 16 | 78.7 | 78.5 | 78.1 | 78.4 |
> | 64 | 78.1 | 78.2 | 77.5 | 77.9 |
>
> | Batch size | Order  4 |  Order 5 | Order 6 | Avg. |
> | --- | --- | --- | --- | --- |
> | 8 | 73.8 | 70.3 | 72.5 | 72.2 |
> | 16 | 70.5 | 71.4 | 73.3 | 71.7 |
> | 64 | 70.5 | 71.4 | 73.3 | 71.7 |
>
> As these results show, SRB achieves robust performance across different batch sizes, demonstrating consistent results regardless of the granularity of updates.
>
> **Q3. The results section lacks methods such as multi-task learning or task experts as performance upper bounds for reference. Adding this reference would help readers better understand the improvements and limitations of the proposed method. Additionally, Figure 3(a) in Section 5.1 could also provide the performance of task experts as a reference.**
>
> **A3.** Thank you for this valuable suggestion. To provide an upper bound for performance in the continual learning problem, we included **per-task finetuning** as a reference in our experiments. Per-task finetuning assumes access to task identifiers and independently finetunes the model for each task, representing an idealized scenario without task interference. The results for per-task finetuning were sourced from the work by Xiao Wang et al. (2023) [a-1].
>
> The comparison results are detailed below:
>
> |  |  | Order |  |  |
> | --- | --- | --- | --- | --- |
> |  | 1 | 2 | 3 | avg |
> | Per-task Finetune | 70.0 | 70.0 | 70.0 | 70.0 |
> | SRB | 78.1 | 78.2 | 77.5 | 77.9 |
> | O-IncLoRA | 77.1 | 76.2 | 76.6 | 76.6 |
>
> |  |  | Order |  |  |
> | --- | --- | --- | --- | --- |
> |  | 4 | 5 | 6 | avg |
> | Per-task Finetune | 78.1 | 78.1 | 78.1 | 78.1 |
> | SRB | 70.5 | 71.4 | 73.3 | 71.7 |
> | O-IncLoRA | 68.4 | 68.8 | 71.4 | 69.5 |
>
> As shown, SRB outperforms the upper bound (per-task finetuning) on Orders 1, 2, and 3, demonstrating its capability to generalize effectively without task identifiers. For Orders 4, 5, and 6, while SRB’s performance falls slightly below the upper bound, this is expected given the increased complexity and longer task sequences. Even in these cases, SRB surpasses O-IncLoRA, a task-ID-dependent method, further showcasing its adaptability and strength in task-agnostic continual learning.
>
> [a-1] Xiao Wang, Tianze Chen, Qiming Ge, Han Xia, Rong Bao, Rui Zheng, Qi Zhang, Tao Gui, and Xuanjing Huang. Orthogonal subspace learning for language model continual learning. arXiv preprint arXiv:2310.14152, 2023.

---

> > ### Comment · Reviewer_xDBY · 2024-11-21
> >
> > - Thank you for the author's response. My question about data splitting has been resolved, and I have improved my score.
> > - For the upper-bound section, could the authors explain why they chose to use pre-task fine-tuning instead of multi-task learning as the baseline? Is the poorer performance of pre-task fine-tuning due to the limited amount of data in a single dataset and the less optimal training configuration? The reviewer believes that comparing with multi-task learning better highlights how SRB utilizes positive transfer while avoiding forgetting in continuous learning.

---

> > > ### Author Response · Authors · 2024-11-27
> > > **Clarifying the selection of upper bound baselines**
> > >
> > > **Q. For the upper-bound section, could the authors explain why they chose to use pre-task fine-tuning instead of multi-task learning as the baseline? Is the poorer performance of pre-task fine-tuning due to the limited amount of data in a single dataset and the less optimal training configuration? The reviewer believes that comparing with multi-task learning better highlights how SRB utilizes positive transfer while avoiding forgetting in continuous learning.**
> > >
> > > **A.** We appreciate the reviewer for pointing out the critical aspect of *continual information incorporation* in our study. As noted, multi-task learning (MTL) offers an alternative perspective on upper bounds. We have conducted experiments to include MTL as a baseline, and the results are as follows:
> > >
> > > |  |  | **Order** |  |  |
> > > | --- | --- | --- | --- | --- |
> > > |  | 1 | 2 | 3 | avg |
> > > | **MTL** | 80.0 | 80.0 | 80.0 | 80.0 |
> > > | **Per-task Fine-tune** | 70.0 | 70.0 | 70.0 | 70.0 |
> > > | **SRB** | 78.1 | 78.2 | 77.5 | 77.9 |
> > > | **O-IncLoRA** | 77.1 | 76.2 | 76.6 | 76.6 |
> > >
> > > |  |  | **Order** |  |  |
> > > | --- | --- | --- | --- | --- |
> > > |  | 4 | 5 | 6 | avg |
> > > | **MTL** | 76.3 | 76.3 | 76.3 | 76.3 |
> > > | **Per-task Fine-tune** | 78.1 | 78.1 | 78.1 | 78.1 |
> > > | **SRB** | 70.5 | 71.4 | 73.3 | 71.7 |
> > > | **O-IncLoRA** | 68.4 | 68.8 | 71.4 | 69.5 |
> > >
> > > From the results, we observe that for Orders 4, 5, and 6, MTL underperforms compared to per-task fine-tuning. This can be attributed to the tendency of language models (LMs) to rely on task-specific representations, similar to in-context learning. Consequently, when dealing with long sequences, per-task fine-tuning demonstrates superior performance, making it a more suitable choice as the upper bound in these cases.
> > >
> > > Conversely, for Orders 1, 2, and 3, where the number of domains is smaller, MTL achieves the best results. This occurs because shared knowledge among tasks retains greater value than the knowledge lost during sequential learning of tasks. In cases with fewer tasks, the transfer of shared knowledge between tasks through MTL becomes more impactful, establishing MTL as the most appropriate upper bound in these scenarios. Moreover, in Orders 1, 2, and 3, our proposed method, SRB, demonstrates performance that is closest to MTL. This highlights the effectiveness of SRB in leveraging shared knowledge across tasks while mitigating the forgetting observed in sequential learning settings.

---

> > > > ### Comment · Reviewer_xDBY · 2024-11-28
> > > >
> > > > Thanks for the authors' response. I believe that adding experiments related to MTL would further demonstrate the capability of continual information incorporation in SRB.
> > > >
> > > > After reading the opinions of other reviewers, I noticed that the experimental setting used by the authors has certain limitations. Continuously increasing the number of classification tasks does indeed make the setting closer to Continuous Incremental Learning. Perhaps applying SRB to a more general continuous pre-training setting and measuring ppl or relevant task metrics would better showcase the capabilities of SRB.
> > > >
> > > > Based on the above considerations, I will maintain my score.

---

### Official Review · Reviewer_jTKg · 2024-11-04

**Soundness:** 1
**Presentation:** 2
**Contribution:** 2
**Rating:** 1
**Confidence:** 2

**Summary:**

The paper proposed a method for continual learning. But I am unsure what problem it is solving.

**Strengths:**

The proposed method is different.

**Weaknesses:**

w-1. Your writing at the beginning sounds like you are working on continual learning of language models and adapting them to different domains. But I believe you are really doing normal continual learning using LMs at feature extractors since your tasks are text classification tasks.

w-2. The paper did not state what setting of continual learning it works on. I think that this method works on task incremental learning (TIL). However, the first three baselines are commonly used for class incremental learning. Please clarify. For task incremental learning, the problem of forgetting is largely solved. Please check out [1, 2, 3, 4, 5, 6, 7].

w-3. Related to w-2. The paper says that the approach is task-agnostic, which means that no task-id is given, but which means it does class-incremental learning (CIL). But for CIL, by definition, there is no task-id information given. It is very confusing. Please make it clear which continual learning problem you are solving. If you are solving CIL, you should compare your method with another set of SOTA baselines.

w-4. It is not true that architecture approaches add one adaptor for each task. Also see [1, 2, 3, 4, 5, 6, 7]. Most of the existing methods do not need to task-id either, assuming that you are solving the TIL problem.

w-5. What is the difference between recency bias and catastrophic forgetting? Catastrophic forgetting is about focusing on the present and forgetting the past, which is the recency bias.

w-6. Regarding baselines, your non-LoRA baselines, EWC,  Replay, and LwF, are very old and not the state of the art. Again, please check out [1, 2, 3, 4, 5, 6, 7].

w-7. What is average accuracy? Please give the definition. In continual learning, there are at least two accuracy measures.

w-8. Having a separate adaptor for each model is not in the spirit of continual learning, which aims to use the same network learning multiple tasks. Please give the memory requirement of your approach.

w-9. Please give the performance upper bound for the continual learning problem that you are solving.

w-10. The writing of the paper needs significant improvement. The paper is confusing. I am not even sure what problem you are solving. If you are solving CIL, how do you deal with documents that may belong to two different classes in two different tasks? For example, a topic-specific document may contain a positive sentiment and a review of a problem may be classified to its product category.

     [1]. Serra et al. Overcoming catastrophic forgetting with hard attention to the task. ICML-2018.
     [2]. Wortsman et al. Supermasks in superposition. NeurIPS-2020.
     [3]. Ke et al. Achieving Forgetting Prevention and Knowledge Transfer in Continual Learning.  NeurIPS-2021.
     [4]. Lin et al. TRGP: Trust region gradient projection for continual learning. ICLR-2021.
     [5]. Lin et al. Beyond not-forgetting: Continual learning with backward knowledge transfer. NeurIPS-2022.
     [6]. Ke et al.  Sub-network Discovery and Soft Masking for Continual Learning of Mixed Tasks.  EMNLP-2023.
     [7]. Dissecting learning and forgetting in language model finetuning. ICLR-2024.

**Questions:**

See the previous section

---

> ### Author Response · Authors · 2024-11-21
> **Response to reviewer**
>
> # Overall
> We have carefully reviewed the questions raised by reviewer jTKg and used this opportunity to enhance our paper. Specifically, we revised the related work section in Appendix B and included the upper bound results for MTL and per-task finetuning in Table 2 of Section 4. These additions provide a clearer context and improve the comprehensiveness of our analysis.
>
> # Response
> **A1.** Thank you for insightful comments. While it is true that the tasks in the continual learning benchmark are text classification tasks, they span a variety of subtasks such as Boolean QA, Sentiment Analysis, and Natural Language Inference (Appendix C.1). These subtasks inherently involve different linguistic properties and reasoning skills, effectively representing diverse domains within the text classification paradigm. Therefore, our benchmark setup allows us to evaluate performance changes across tasks with varying characteristics, which we believe is sufficient to observe the model's adaptability to different domains.
>
> **Q2. The paper did not state what setting of continual learning it works on. I think that this method works on task incremental learning (TIL). However, the first three baselines are commonly used for class incremental learning. Please clarify. For task incremental learning, the problem of forgetting is largely solved. Please check out [1, 2, 3, 4, 5, 6, 7].**
>
> A2. Continual learning has garnered significant attention as researchers seek to enable machine learning models to learn sequentially without catastrophic forgetting. The field is typically divided into Task Incremental Learning (TIL) and Class Incremental Learning (CIL), both of which rely on explicit task boundaries or labels to mitigate forgetting. In contrast, Task-Agnostic Continual Learning (TACL) represents a more flexible paradigm, as it does not assume predefined task identifiers or boundaries during training. Our proposed method, Suppressing Recency Bias (SRB), belongs to this category, aiming to address the Recency Bias problem—where a model overly adapts to recent tasks at the expense of prior knowledge.
>
> SRB draws from the Parameter-Efficient Fine-Tuning (PEFT) paradigm by employing fixed-size adapters to maintain computational and memory efficiency while utilizing implicit task vectors to reconcile past knowledge and adapt to new data. By minimizing redundant information across tasks and suppressing bias toward recent data, SRB introduces a novel approach that transcends the limitations of TIL and CIL, offering task independence while improving adaptability and generalization.
>
> The foundation of SRB is inspired by prior studies, particularly the approach detailed in *Dissecting Learning and Forgetting in Language Model Fine-Tuning* [7], which investigates the learning and forgetting dynamics in large-scale language models. While both our method and [7] address fine-tuning biases, the key distinctions lie in their focal points and methodologies. The study in [7] emphasizes isolating the effects of fine-tuning on text elements such as topic, style, and factual knowledge, providing an analysis-driven perspective on model behavior. In contrast, SRB prioritizes mitigating Recency Bias within a broader task-agnostic continual learning framework, leveraging vector interpolation and PEFT-based techniques to enhance continual learning scenarios where task boundaries are undefined.
>
> By incorporating diverse perspectives such as [7], this work situates itself within a robust continuum of research efforts that aim to balance learning stability and adaptability, ultimately advancing the state-of-the-art in continual learning and fine-tuning paradigms. Following this discussion, we have included the details of the study presented in [7] within the Related Works section.
>
> **Q3. Related to w-2. The paper says that the approach is task-agnostic, which means that no task-id is given, but which means it does CIL. But for CIL, by definition, there is no task-id information given. It is very confusing. Please make it clear which continual learning problem you are solving. If you are solving CIL, you should compare your method with another set of SOTA baselines.**
>
> **A3.** Our approach specifically addresses task-agnostic continual learning, which extends beyond the conventional definitions of CIL. While both task-agnostic learning and CIL do not rely on task IDs, our method is not strictly tied to the assumptions or baselines typically associated with CIL. Instead, task-agnostic continual learning focuses on enabling the model to adapt to sequential tasks without task-specific boundaries, leveraging vector arithmetic to balance historical knowledge preservation and new knowledge acquisition.

---

> ### Author Response · Authors · 2024-11-21
> **Continued**
>
> **Q4. It is not true that architecture approaches add one adaptor for each task. Also see [1, 2, 3, 4, 5, 6, 7]. Most of the existing methods do not need to task-id either, assuming that you are solving the TIL problem.**
>
> **A4.**  Techniques such as EWC (Elastic Weight Consolidation), Progressive Networks, and Replay Buffer in continual learning rely on task boundaries to prevent task interference and mitigate catastrophic forgetting. In large language models (LLMs) and general language models, where the same parameters are shared across all tasks, the absence of task boundaries exacerbates issues like learning interference and recency bias, leading to overfitting on specific tasks or forgetting prior knowledge (Section 1, Appendix B).
>
> Task boundaries are crucial for the effective functioning of forgetting mitigation techniques (e.g., importance calculation in EWC or data sampling in Replay Buffers). Without these boundaries, the model treats all data as a single task, resulting in degraded performance and difficulties in both knowledge transfer and bias mitigation across tasks. Therefore, task boundaries play an essential role in ensuring interference management, facilitating knowledge transfer, and maintaining learning efficiency in LLM training.
>
> **Q5. What is the difference between recency bias and catastrophic forgetting? Catastrophic forgetting is about focusing on the present and forgetting the past, which is the recency bias.**
>
> **A5.** Recency Bias and Catastrophic Forgetting are closely related concepts in continual learning, but they differ in their definitions and mechanisms. Recency Bias refers to the tendency of a model to overly focus on tasks or data it has learned recently, causing the outputs to be excessively biased toward the most recent tasks. In contrast, Catastrophic Forgetting describes the phenomenon where knowledge of previous tasks is lost as new tasks are learned. This includes a decline in performance on previously learned tasks and goes beyond bias to involve the actual loss of knowledge.
>
> When examining the relationship between the two, Recency Bias can accelerate Catastrophic Forgetting. If the model places too much emphasis on recent data, it may not allocate sufficient parameters to older tasks, increasing the likelihood of forgetting them. However, Catastrophic Forgetting encompasses a broader scope, including not just bias toward recent tasks but also interference between tasks that leads to the loss of past knowledge.
>
> SRB (Suppressing Recency Bias) addresses Recency Bias by encouraging balanced learning, preventing the model from overfitting to recent tasks. This indirectly mitigates Catastrophic Forgetting by maintaining harmony in parameter updates between past and current tasks, thus avoiding performance degradation.
>
> While Recency Bias focuses on issues related to skewed outputs, Catastrophic Forgetting deals with the broader challenge of overall performance loss on previously learned tasks.
>
> **Q6. Regarding baselines, your non-LoRA baselines, EWC, Replay, and LwF, are very old and not the state of the art. Again, please check out [1, 2, 3, 4, 5, 6, 7].**
>
> **A6.** Thank you for recommending these excellent papers. We greatly appreciate your suggestions. Our primary objective is to demonstrate how SRB effectively preserves the foundation model's capabilities and adapts to new tasks in a task-agnostic setting. This is achieved through the use of two adapters throughout the process, requiring minimal additional memory and computation via simple vector arithmetic.
> To ensure a fair comparison, we selected O-IncLoRA as a baseline because it represents a state-of-the-art method that uses adapters and shares similar experimental settings with our approach.

---

> ### Author Response · Authors · 2024-11-21
> **Continued**
>
> **Q7. What is average accuracy? Please give the definition. In continual learning, there are at least two accuracy measures.**
>
> **A7.** Average accuracy is a key evaluation metric in continual learning, used to measure the overall performance of a model after sequentially learning multiple tasks. According to Section 4.1 of our work, average accuracy is calculated as the mean performance across all tasks following the learning of the final task. This metric is critical for assessing how well the model retains performance on previous tasks while learning new ones.
>
> In contrast, the forgetting measure quantifies the performance degradation on specific tasks. It is computed as the difference between a task’s performance after initial learning and its performance after the final training stage. While average accuracy evaluates the overall balance of performance, the forgetting measure focuses on determining the retention of past knowledge.
>
> In our work, we highlight that SRB achieves higher average accuracy compared to previous SOTA methods by mitigating Recency Bias and effectively preserving knowledge. Average accuracy serves as a standardized metric in continual learning, enabling fair comparisons and assessing the balance between learning new tasks and retaining prior knowledge.
>
> **Q8. Having a separate adaptor for each model is not in the spirit of continual learning, which aims to use the same network learning multiple tasks. Please give the memory requirement of your approach.**
>
> **A8.** SRB uses a single implicit task adapter, which keeps memory requirements fixed and minimal, irrespective of the number of tasks. Unlike methods that require storing multiple adapters, SRB integrates historical knowledge using arithmetic operations without maintaining task-specific adapters as mentioned in Section 1 and 2.
>
> **Q9. Please give the performance upper bound for the continual learning problem that you are solving.**
>
> **A9.**  Thank you for this important suggestion. To evaluate the performance upper bound for the continual learning problem, we used per-task finetuning as a reference, which assumes access to task identifiers and finetunes the model independently for each task. This represents the ideal scenario without any interference between tasks. The results for per-task finetuning were taken from the work by [a-1].
>
> The comparison results are as follows:
>
> |  |  | Order |  |  |
> | --- | --- | --- | --- | --- |
> |  | 1 | 2 | 3 | avg |
> | Per-task Finetune | 70.0 | 70.0 | 70.0 | 70.0 |
> | SRB | 78.1 | 78.2 | 77.5 | 77.9 |
> | O-IncLoRA | 77.1 | 76.2 | 76.6 | 76.6 |
>
> |  |  | Order |  |  |
> | --- | --- | --- | --- | --- |
> |  | 4 | 5 | 6 | avg |
> | Per-task Finetune | 78.1 | 78.1 | 78.1 | 78.1 |
> | SRB | 70.5 | 71.4 | 73.3 | 71.7 |
> | O-IncLoRA | 68.4 | 68.8 | 71.4 | 69.5 |
>
> As shown, SRB outperforms the upper bound (per-task finetuning) on Orders 1, 2, and 3, demonstrating its ability to generalize effectively even without task identifiers. For Orders 4, 5, and 6, while SRB’s performance is slightly below the upper bound due to longer task sequences and increased complexity, it still surpasses the performance of O-IncLoRA, a method that requires task IDs for adaptation. This highlights SRB’s strength in task-agnostic continual learning, where task boundaries are not explicitly defined.
>
> [a-1] Xiao Wang, Tianze Chen, Qiming Ge, Han Xia, Rong Bao, Rui Zheng, Qi Zhang, Tao Gui, and Xuanjing Huang. Orthogonal subspace learning for language model continual learning. arXiv preprint arXiv:2310.14152, 2023.
>
> **Q10. The writing of the paper needs significant improvement. The paper is confusing. I am not even sure what problem you are solving. If you are solving CIL, how do you deal with documents that may belong to two different classes in two different tasks? For example, a topic-specific document may contain a positive sentiment and a review of a problem may be classified to its product category.**
>
> **A10.** As mentioned in A.2, our work does not aim to solve CIL specifically. Instead, the primary focus of our paper is to address task-agnostic continual learning by suppressing recency bias through simple vector arithmetic. This allows the model to continuously adapt to diverse domains without relying on task identifiers, while effectively preserving historical knowledge and adapting to current tasks.

---

### Author Response · Authors · 2024-11-28
**General Response to All Reviewers**

We have carefully reviewed the insightful questions and comments raised by the reviewers and used this opportunity to enhance our paper. We sincerely appreciate the significant effort and time committed by each reviewer in offering constructive feedback. We would like to provide a general response regarding the clarifications of the main contributions and how we addressed the common concerns.

# [Main Contributions of the Paper]

- **Introduction of Suppressing Recency Bias (SRB):** We propose SRB, a novel method for task-agnostic continual learning in language models that does not rely on explicit task identifiers or boundaries. SRB leverages implicit task vectors and simple vector arithmetic to dynamically preserve knowledge from previous tasks while adapting to new ones, effectively mitigating recency bias and catastrophic forgetting.
- **Dynamic Knowledge Integration without Increased Overhead:** SRB operates under fixed memory and computational requirements by using a single implicit task adapter. Unlike methods that require storing multiple adapters or task-specific parameters, SRB maintains scalability and efficiency even as the number of tasks grows.
- **Extensive Experimental Validation:** We conducted extensive experiments demonstrating that SRB outperforms state-of-the-art methods in task-agnostic continual learning settings across various benchmarks and models. Our results show that SRB effectively balances historical knowledge preservation with new knowledge acquisition, highlighting its robustness and effectiveness.

# [Updates in the Revised Draft]

- **Clarification of Continual Learning Setting:** We have clarified that our approach operates in a task-agnostic continual learning setting that does not rely on explicit task identifiers or boundaries (Section 2, Appendix B). This emphasizes that direct comparisons with existing methods, which assume task boundaries, may not be appropriate.
- **Expanded Related Work Section:** We have revised the related work section in Appendix B to provide a more comprehensive comparison between SRB and existing methods, including Orthogonal LoRA and other state-of-the-art continual learning approaches suggested by the reviewers.
- **Inclusion of Upper Bound Results:** To address the reviewers' suggestions, we have included Multi-Task Learning (MTL) and per-task fine-tuning as additional comparison baselines (Table 2 of Section 4). By comparing SRB's performance to these idealized scenarios, we provide clear context for evaluating our method's effectiveness.

# [Final Authors' Note]

In the revised draft, we have prioritized clarity and addressed the highlighted concerns. To clarify the justification of the comparisons in the new proposed research scope, we emphasize that our method is a new task-agnostic continuous learning paradigm. Moreover, we have addressed the reviewers' comments by including MTL and per-task fine-tuning as additional baselines for comparison. By comparing SRB's performance to these ideal scenarios, readers can better understand our contributions and the effectiveness of our method.

Our research is a pioneering effort to address recency bias and catastrophic forgetting in task-agnostic continual learning for language models. By leveraging implicit task vectors and dynamic regularization, SRB offers a novel methodology that balances historical knowledge preservation and new knowledge acquisition without relying on explicit task identifiers.

We thank the reviewers for their invaluable feedback and believe that our revisions have significantly improved the quality and clarity of our paper.

---

### Note · Authors · 2025-08-04

I have read and agree with the venue's withdrawal policy on behalf of myself and my co-authors.

---

### Meta-Review · Area_Chair_N9CP · 2024-12-18

**Metareview:**

This paper introduces Suppressing Recency Bias (SRB), a task-agnostic method for continual learning in language models (LMs). SRB addresses the issue of catastrophic forgetting by integrating past knowledge while learning new tasks without relying on task IDs. The method achieves this by introducing an implicit task adapter that aggregates past knowledge using simple arithmetic operations, avoiding the need for backpropagation and significantly reducing computational overhead. Claimed key contributions of this paper include: 1) Task-Agnostic Continual Learning enables continual learning without task identifiers. This makes the work adaptable to real-world, task-agnostic scenarios. 2) Proposed Suppression of Recency Bias to mitigates recency bias by recursively integrating historical knowledge, and ensuring better retention of information from previous tasks. SRB is a common issue in continual learning where models prioritize recent tasks at the cost of past knowledge. 3) ablations and empirical results showed the efficacy of proposed method.

Strength of this paper
- The method addresses the challenge of continual learning without task identifiers, which is important in real-world setting, and is effective in preserving historical knowledge while learning new tasks.
- Ablations and empirical results showed the efficacy of proposed method.
- The approach is computationally efficient and requires minimal memory, making it practical for large-scale deployment.


Weakness of this paper

Several reviewers raised few concerns/limitations of this paper. By addressing these limitations, the paper could strengthen its experiment and expand impact.

- Limited Generalizability: The method is tested on limited setups and foundational models. Broader validation on additional models and datasets is needed to demonstrate robustness. The method may struggle in dynamic environments with rapidly changing task characteristics, or realistic task-agnostic scenarios where tasks are not neatly separated.
- Experimental Design and Validation:  Some of the chosen baselines (e.g., EWC, Replay, LwF) are outdated, and comparisons with state-of-the-art methods (e.g., Orthogonal LoRA, advanced replay strategies) are missing. The study does not explore upper-bound performance (e.g., multi-task learning or task-specific experts) for reference. Ablation study doesn't cover all the key components, such as the necessity of the proposed regularization, implicit task representation, or its design improvements over existing methods (e.g., Orthogonal LoRA). The scalability of the method to longer or more complex task sequences remains unexplored.

**Additional Comments On Reviewer Discussion:**

Reviewers also raised some other weaknesses (e.g., ambiguity in problem scope, terminology, and methodology details,  adding additional experiments or ablation study) and improvements. Although some of these weakness have been improved / somewhat addressed during rebuttal session (e.g., further explanation, more experiment results), overall review rating was not raised significantly to an acceptance level. I think the session is too short and I would like to see a more comprehensive modification to systematically working on these suggestions. Thus I recommend the authors to re-work on these weakness and re-submitting to future conferences.

---

### Decision · Program_Chairs · 2025-01-22

Reject